**Investigation**

# The relationship between sexual dimorphism and intersex correlation: do models support intuition?

Gemma Puixeu (ID) ,* Laura Katherine Hayward (ID) *,†

Institute of Science and Technology Austria, Klosterneuburg 3400, Austria

*Corresponding authors: Gemma Puixeu, Institute of Science and Technology Austria, Klosterneuburg 3400, Austria. Email: gpuixeus@ist.ac.at; Laura Katherine Hayward, Institute of Science and Technology Austria, Klosterneuburg 3400, Austria. Email: laura.hayward@ist.ac.at.
†Senior author.

The evolution of sexual dimorphism ($SD$) (the difference in average trait values between females and males) is often thought to be constrained by shared genetic architecture between the sexes. Indeed, it is commonly expected that $SD$ should negatively correlate with the intersex correlation (the genetic correlation between effects of segregating variants in females and males, $r_{fm}$), either because (1) traits with ancestrally low $r_{fm}$ are less constrained in their ability to respond to sex-specific selection and thus evolve to be more dimorphic, or because (2) sex-specific selection, driving sexual dimorphism evolution, also acts to reduce $r_{fm}$. Despite the intuitive appeal and prominence of these ideas, their generality and the conditions in which they hold remain unclear. Here, we develop models incorporating sex-specific stabilizing selection, mutation, and genetic drift to examine the relationship between $r_{fm}$ and $SD$. We show that the two commonly-discussed mechanisms with the potential to generate a negative correlation between $SD$ and $r_{fm}$ could just as easily generate a *positive* association, since the standard line of reasoning hinges on a hidden assumption that sex-specific adaptation more frequently favors increased dimorphism than reduced dimorphism. Our results provide, to our knowledge, the first mechanistic framework for understanding the conditions under which a correlation between $r_{fm}$ and $SD$ may arise and offer a compelling explanation for inconsistent empirical evidence. We also make the intriguing observation that—even when selection between the two sexes is identical—drift generates nonzero $SD$. We quantify this effect and discuss its significance.

Keywords: sexual dimorphism; intersex correlation; genetic drift; sex-specific selection; polygenic adaptation; infinitesimal limit; quantitative genetics; breeder's equation

## Introduction

Females and males are often subject to unequal selective pressures arising from divergent ecological niches and reproductive interests, leading to distinct optimal trait values. These differences typically drive the evolution of sexual dimorphism ($SD$), corresponding to the difference in mean trait values between females and males (Rice and Chippindale 2001; see Box 1 for some definitions). However, this evolution is limited by the fact that, even in those species with sex chromosomes, the two sexes share the vast majority of their genome (Bachtrog et al. 2014). Consequently, the establishment of sex differences typically relies on the decoupling of the genotype-to-phenotype relationship between the sexes, i.e. it requires at least some new mutations affecting the trait to have different effects in females and males (Mank 2017).

From a quantitative genetics perspective, the extent to which the genetic architecture in a trait is shared between the sexes is typically measured by the intersex correlation ($r_{fm}$; Lande 1980). $r_{fm}$ is the genetic correlation between effects on the trait of segregating variants in females and males, and it can be empirically estimated by comparing sex-specific phenotypes in breeding designs of known relatedness between individuals (e.g. Bonduriansky and Rowe 2005). A high $r_{fm}$ implies that segregating variants exert similar effects on sisters and brothers, whereas a low $r_{fm}$ suggests that a variant increasing a sister's trait value could easily reduce that of her brother. Intersex correlation is therefore considered a key predictor of how populations respond to sex-specific selection, and its impact on the evolution of sexual dimorphism has been extensively discussed in the field of sex-specific adaptation.

Concretely, it is often assumed that intersex correlation and sexual dimorphism should negatively correlate with one another (Lande 1980, 1987; Bonduriansky and Rowe 2005; Fairbairn 2007; Poissant et al. 2010; Stewart et al. 2010). Two hypotheses are most commonly provided as potential explanations (stated in e.g. Bonduriansky and Rowe 2005; Fairbairn 2007; Griffin et al. 2013; Stewart and Rice 2018; McGlothlin et al. 2019): first, that traits with ancestrally low $r_{fm}$ are less constrained in their ability to respond to sex-specific selection and thus evolve to be more dimorphic; second, that sex-specific selection (which leads to the evolution of sexual dimorphism) acts to reduce the $r_{fm}$.

In line with the first hypothesis (discussed, for example, in Bolnick and Doebeli 2003; Poissant et al. 2010; Stewart et al. 2010) is the idea that sexual dimorphism will easily (hardly) evolve for traits with a low (high) intersex correlation (Stewart et al. 2010; Stewart and Rice 2018). The potential for a high intersex correlation to pose a long-term constraint on the evolution of sex differences has been illustrated by some artificial selection

**Box 1: Terminology**

- **Intersex correlation**: ratio between intersex covariance and geometric mean of sex-specific averages (Equation (3)). It measures the correlation between the additive effects of segregating variants as expressed in females and males.
- **Sexual dimorphism**: absolute value of the difference between female and male trait means (Equation (6)). It reflects the magnitude of the difference between sex-specific averages.
- **Signed sexual dimorphism**: difference between female and male trait means (Equation (7)). It reflects the magnitude and direction of sexual dimorphism.
- **Concordant adaptation**: dynamics of sex-specific trait means after a change in the average of sex-specific trait optima. Adaptation is purely concordant after a shift in optima of equal magnitude and direction between the sexes. When we refer to concordant adaptation we typically mean purely concordant.
- **Discordant adaptation**: dynamics of sex-specific trait means after a change in the difference between sex-specific trait optima. Adaptation is purely discordant after a shift in optima of equal magnitude and opposite direction between the sexes. When we refer to discordant adaptation we typically mean purely discordant. There are two types of discordant shifts:
  - **Divergent shifts** bring sex-specific optima farther apart
  - **Convergent shifts** bring sex-specific optima closer together

experiments (Harrison 1953; Reeve and Fairbairn 1996; Stewart and Rice 2018). Most notably, Stewart and Rice (2018) observed a minimal change in sexual dimorphism in fly body size after as many as 250 generations of selection for sexual dimorphism. However, multiple studies have also provided evidence for fast, seemingly unconstrained, evolution of sexual dimorphism (Frankham 1968a, 1968b; Bird and Schaffer 1972; Eisen and Hanrahan 1972; Zwaan et al. 2008; Delph et al. 2011; Kaufmann et al. 2021). For example, Bird and Schaffer (1972) selected fruit flies for sexual dimorphism on wing size and found a significant change in sex differences after only 15 generations. Although many of these empirical studies relied on selection following family-based selection designs, unlikely to occur in nature, the qualitative differences in their outcomes are usually attributed to differences in genetic architecture underlying those traits. Specifically, that traits with a high (low) intersex correlation easily (hardly) decouple between the sexes (Stewart et al. 2010).

The prediction that high $r_{fm}$ constrains sexual dimorphism evolution is supported by models of sex-specific adaptation of quantitative traits, first formulated by Lande (1980), who showed that intersex correlation determines the rate of sexually-discordant adaptation (adaptation in response to a change in the *difference* between sex-specific optima; see Box 1 for a more detailed explanation). Nevertheless, from the same models, it follows that as long as intersex correlation is imperfect ($r_{fm} < 1$) and given enough time, sexual conflict will be fully resolved. This suggests that, while $r_{fm}$ poses a constraint on the *speed* of sex-specific adaptation, it is not predictive of the extent of sexual dimorphism eventually achieved. Most two-sex models of this process (e.g. Lande 1980; Cheverud et al. 1985) have assumed an infinitesimal genetic architecture (Lande 1976; Barton et al. 2017), which ignores individual loci and assumes that genic (co)variances remain constant over time. However, we know that considering different genetic architectures can lead to qualitatively different results (as discussed in e.g. Rhen 2000; Reeve and Fairbairn 2001). For example,

in single-locus (or, more generally, genetic variance-limited) models of sex-specific selection, sexual conflict is not resolved unless the locus can evolve to have sex-specific effects (Kidwell et al. 1977; Rice 1984; Rhen 2000; Morrow and Connallon 2013), and more realistic models considering polygenic genetic architectures (Reeve and Fairbairn 2001; Muralidhar and Coop 2024) involve changes in genetic (co)variances over time, and thus display phenotypic dynamics that deviate from the infinitesimal predictions. In general, the relationship between sexual dimorphism and intersex correlation with a polygenic genetic architecture remains largely uncharacterized.

The second hypothesis states that a negative relationship between intersex correlation and sex differences arises because sex-specific selection favors genetic modifications that reduce the intersex covariance, which allows sex-specific adaptation (Lande 1980, 1987; Bondwith and Rowe 2005; Bondwith and Chenoweth 2009; McGlothlin et al. 2019). Indeed, according to the standard picture of sexual dimorphism evolution (as discussed in e.g. Rice and Chippindale 2001; Bondith and Rowe 2005; Cox and Calsbeek 2009; Morrow 2015), an initially monomorphic trait that becomes subject to sex-specific selection will decouple between sexes, allowing sex-specific means to approach their optima and resolve sexual conflict. The idea that this process involves a decrease in intersex correlation traces back to Fisher (1958) (Chapter 6) and Lande (1980), who suggested that genes with sex-limited effects would accumulate over time leading to the prediction that $r_{fm}$ will decrease as sexual dimorphism evolves. However, neither author presented a mathematical justification for this suggestion. Instead, it seems to be based on an intuition of how the intersex correlations should evolve, potentially implying the evolution of sex-specific modifiers, and generally an evolving genetic architecture (Bondith and Rowe 2005), allowing for a stable, long-term reduction in intersex correlation (Bondith and Rowe 2005; Williams and Carroll 2009; Stewart et al. 2010). Nevertheless, the evolution of genetic architecture in general (e.g. for traits with shared genetic bases, like allometric traits; Jones et al. 2003; Barker et al. 2010; Rajon and Plotkin 2013; Yamamichi 2022) and in the context of sexual dimorphism (Williams and Carroll 2009; Stewart et al. 2010) is likely to be a very slow process. As such, changes in the genetic architecture underlying sex-specific trait expression are probably not occurring within the scope of shorter-term evolutionary processes, including most artificial selection experiments cited above, where phenotypes evolve without major changes in genetic architecture.

The two common hypotheses, together with the pattern they are believed to generate, seem intuitive. However, despite their prominence in discussions of the joint evolutionary dynamics of intersex correlation and sexual dimorphism in the context of sex-specific adaptation, their universality remains unestablished, and the underlying mechanisms and assumptions are poorly understood. On the one hand, empirical evidence is inconsistent: while several studies suggest that greater sexual dimorphism correlates with lower $r_{fm}$ across traits and species (e.g. Delph et al. 2004, 2010; Bondith and Rowe 2005; McDaniel 2005; Fairbairn 2007; Poissant et al. 2010; Griffin et al. 2013; Cox et al. 2017), many findings are only marginally significant, and other studies fail to detect a significant association (Cowley and Atchley 1988; Preziosi and Roff 1998; Chenoweth and Blows 2003; Ashman and Majetic 2006; Leinonen et al. 2011; Puixeu et al. 2019). This, in spite of the expectations described above, speaks against the universality of such a pattern. On the other hand, theoretical work, providing a mechanistic understanding of the conditions in which this

negative association is expected, is similarly sparse. Existing studies largely rely on verbal predictions (Lande 1980, 1987), focus on within-generation change in $r_{fm}$ with no explicit model for its evolution (Barker et al. 2010; McGlothlin et al. 2019), or draw conclusions based solely on simulation results (Reeve and Fairbairn 2001). Addressing this gap in understanding of the co-evolutionary dynamics of sexual dimorphism and intersex correlation is the main motivation of the current study.

We formulate a model of sex-specific stabilizing selection, mutation, and drift (a two-sex extension of Hayward and Sella 2022), which is a common regime in sex-specific adaptation (Prasad et al. 2007; Abbott et al. 2010; Stulp et al. 2012; Sanjak et al. 2018), and analyze the sex-specific evolutionary dynamics after a shift in sex-specific optima, while keeping track of intersex correlation over time. Given that the dynamics seem to strongly depend on the assumptions on the genetic architecture, we compare the predictions of the deterministic infinitesimal model with the evolutionary outcomes of simulations considering two types of highly polygenic architectures. The first is an approximately infinitesimal architecture, where all contributing alleles have small effect sizes and do not experience substantial changes in frequency under directional selection. The second is a less infinitesimal architecture with a significant proportion of large-effect mutations, which in humans seems to be the genetic architecture underlying most complex traits, as suggested by numerous genome-wide association studies (GWAS; e.g. Wood et al. 2014; Locke et al. 2015; Simons et al. 2018).

We consider these genetic architectures to be non-evolving (i.e. we are not considering modifier loci that could lead to stable decreases in intersex covariances). While this likely excludes certain mechanisms that might contribute to stable reductions in $r_{fm}$ during sexual dimorphism evolution, as suggested by the second hypothesis above, we make this choice for four reasons. First and most importantly, it is the natural first step: we cannot hope to understand the relationship between intersex correlation and sexual dimorphism in the most general setting without first understanding their co-evolutionary dynamics with a non-evolving genetic architecture. This is particularly important given that some of our findings with a non-evolving architecture are unexpected. Second, the evolution of intersex covariances is expected to be a slow process, so our assumptions are likely to reflect the dynamics of shorter-term evolutionary processes (Williams and Carroll 2009; Stewart et al. 2010). Third, our results are more directly comparable to those of most prior studies, which have also assumed a non-evolving genetic architecture (Lande 1980; Reeve and Fairbairn 2001; Wyman et al. 2013). Fourth, some of our conclusions are expected to be robust to relaxing this assumption (see *Discussion* for more details).

Our results confirm Lande (1980)'s prediction that, at equilibrium under stabilizing selection, intersex correlation is independent of sexual dimorphism in infinitely large populations with deterministic dynamics. By deriving an expression for sexual dimorphism that accounts for the effects of genetic drift, we show that this independence carries over to finite populations. However, we also find that the classical deterministic predictions for sexual dimorphism are not entirely accurate in finite populations. Notably, our results reveal that, even when selection pressures are identical between the sexes, genetic drift generates nonzero sexual dimorphism, with a predictable magnitude. We explicitly quantify this equilibrium dimorphism and discuss its significance.

By considering the transient phase of adaptation to new sex-specific optima (during which directional selection acts), we illustrate that mechanisms underlying the two extensively-discussed

hypotheses to explain a negative association between intersex correlation *can* both generate a relationship between the two, even with a non-evolving genetic architecture. Crucially, however, we show that the association generated is only negative if adaptation more frequently favors increased dimorphism over decreased dimorphism, i.e. if divergent shifts in optima, which increase the distance between sex-specific optima, are more common than convergent shifts which decrease the distance (see Box 1 for a more detailed explanation of the terminology). Indeed, we find that if convergent shifts are more common than divergent shifts the same two mechanisms can generate a *positive* association between sexual dimorphism and intersex correlation. This is important because it exposes a hidden assumption behind the prevailing intuition: namely, that divergent shifts are consistently favored over convergent shifts. To our knowledge, there is no reason to expect that this should be the case.

Additionally, in the course of our investigation into the relationship between sexual dimorphism and intersex correlation, we examine in detail the dynamics of sex-specific adaptation under stabilizing selection, mutation, and drift, with a highly polygenic genetic architecture. Incorporating the effects of genetic drift, we derive novel expressions for sex-specific variances, the covariance between sexes, intersex correlation, and sexual dimorphism at equilibrium. We further analyze how the phenotypic response to a shift in the optimum arises from allele frequency dynamics, extending the framework of Muralidhar and Coop (2024)—which is limited to genetic architectures where predictions from the infinitesimal limit hold—and generalizing the single-sex results of Hayward and Sella (2022). Regarding the response of sex-specific means, we delineate the conditions under which deviations from Lande's classical predictions become appreciable. While previous studies (e.g. Reeve and Fairbairn 2001) have discussed such deviations in terms of changes to (co)variances, we demonstrate that third-order central moments of the phenotypic distribution—which emerge in our generalization of the two-sex breeder's equation— also play a critical role, particularly after the initial rapid phase of adaptation. Finally, we characterize the long-term equilibration process by providing approximations for the rate at which the component of the mean phenotype maintained by fixations, rather than segregating variation, converges to the new optimum—a description, to our knowledge, not previously offered in two-sex models.

Altogether, in this study, we take classical results and well-established expectations about the evolutionary interplay between sexual dimorphism and intersex correlation as the starting point. We re-examine these results from a new perspective, formally articulating the commonly accepted reasoning behind the expectation of a negative correlation between the two. Our analysis challenges prevailing intuition by uncovering the implicit assumptions underlying these arguments, thereby highlighting the importance of clearly stating the assumptions and mechanisms that underpin widely held hypotheses. Moreover, we show how established results integrate into a broader mathematical framework, providing a more complete description of the evolutionary dynamics of a trait under sex-specific stabilizing selection, both at and away from equilibrium.

## Methods
### The model

We define a two-sex extension of the standard model for the evolution of a highly polygenic, quantitative trait under stabilizing selection (Wright 1935; Simons et al. 2018; Hayward and Sella 2022). Assuming additivity, an individual's phenotypic value follows

from its genotype (Lynch and Walsh 1998), and is given, for females ($z_f$) and males ($z_m$), by

$$z_f = \sum_{i=1}^{L} a_{i,f} g_i + \epsilon_f; \quad z_m = \sum_{i=1}^{L} a_{i,m} g_i + \epsilon_m. \tag{1}$$

The first term is the genetic contribution, given by the sum of sex-specific phenotypic effects ($a_{i,f}$ and $a_{i,m}$), with $g_i = 0, 1,$ or $2$ indicating the number of copies of allele $i$ inherited by the individual, and $L$ being the target size of the trait. The second term is the sex-specific environmental contribution, which we take to be normally distributed and independent of the genetic contribution ($\epsilon_\alpha \sim N(0, V_{E,\alpha}$ for $\alpha = f, m$).

Stabilizing selection is modeled via sex-specific Gaussian fitness functions, where fitness declines with distance from sex-specific optima ($O_f, O_m$)

$$
\begin{aligned}
W_f(z_f) &= \mathrm{Exp}\left[-\frac{\gamma_f^2 (z_f - O_f)^2}{V_S}\right] \\
W_m(z_m) &= \mathrm{Exp}\left[-\frac{\gamma_m^2 (z_m - O_m)^2}{V_S}\right].
\end{aligned}
\tag{2}
$$

Here $1/V_S$ determines the overall strength of stabilizing selection; $\gamma_f$ and $\gamma_m$ modulate the proportion of selection that acts on each sex, and satisfy $\gamma_f^2 + \gamma_m^2 = 1$. We assume that neither sex is evolving neutrally, so sex-specific selection strengths, $1/V_{S,f} \equiv 2\gamma_f^2/V_S$ and $1/V_{S,m} \equiv 2\gamma_m^2/V_S$, are nonzero (i.e. $\gamma_f, \gamma_m > 0$). We choose to parameterize the problem in terms of $\gamma_f, \gamma_m,$ and $V_S$ instead of $V_{S,f}, V_{S,m}$ because it allows us to separate the overall strength of selection and the proportion that acts on each sex; however, replacing them with $V_{S,f}, V_{S,m}$ recovers the parameterization used in previous work (e.g. Lande 1980). Since the sex-specific additive environmental contributions to phenotypic variation can be absorbed into $V_{S,f}, V_{S,m}$ (by replacing them with $V'_{S,f} = V_{S,f} + V_{\epsilon,f}; V'_{S,m} = V_{S,m} + V_{\epsilon,m}$, Turelli 1984), we consider only the genetic contributions.

The population evolves according to the standard model of a diploid, panmictic population of constant size $N$, with non-overlapping generations. Exactly half of individuals are female and the other half male and, each generation, mothers and fathers are randomly chosen to reproduce with probabilities proportional to their fitness (via Wright–Fisher sampling with fertility selection). This is followed by mutation, free recombination, and Mendelian segregation. We use the infinite sites approximation, which is accurate provided that the per site mutation rate, $\mu$, is sufficiently low so that very few sites are hit by mutation more than once over relevant timescales ($4N\mu \ll 1$). Consequently, we sample the number of new mutations per gamete per generation from a Poisson distribution with mean $U = L\mu$.

The sex-specific effect sizes of incoming mutations, $a_f$ and $a_m$, are obtained as follows: we draw the overall scaled strength of stabilizing selection on the allele ($2Ns_e$) from an exponential distribution with a specific average (see Simulations section), and we determine the fraction of stabilizing selection that acts on the allele via females (and males) from a second distribution (more details provided in the section on genetic architecture). Sex-specific effect sizes follow from these two quantities (using Equation (15) in the section on genetic architecture). For each mutation, we assume there is an equal probability of it being positive or negative (increasing or decreasing the trait value). In Table 1, we provide a summary of all notation used.

## Parameter ranges and choice of units

We examine the genetic and phenotypic dynamics of a two-sex population adapting to changes in sex-specific optima. We follow previous studies (Simons et al. 2018; Hayward and Sella 2022) in defining the working parameter ranges to ensure that the conditions assumed by the analytic framework hold.

In particular, we assume that the trait is highly polygenic ($2NU \gg 1$) and subject to substantial but not catastrophically strong stabilizing selection. We further assume that the distance between the optimum phenotype in females ($O_f$) and that in males ($O_m$) is not massive relative to the width of the fitness function, i.e. $|O_f - O_m| \lesssim 0.5\sqrt{V_S}$ (where the symbol $\lesssim$ denotes less than or on the same order as); see Supplementary Section 3 for details. Under these assumptions, the phenotypic distribution at stabilizing selection–mutation–drift balance is symmetric, and the sex-specific mean phenotypes exhibit small, rapid fluctuations around the respective optima, with the variance of those fluctuations given by $\delta^2 = V_S/(2N)$ in the infinitesimal limit (Bürger and Lande 1994). The phenotypic variance is greater than these fluctuations $V_A > \delta^2$, but substantially smaller than the width of the fitness function $V_A \ll V_S$.

After ensuring that the population is at equilibrium under mutation–selection–drift balance, we apply a shift in sex-specific optima $\Lambda_f, \Lambda_m$. We assume that the magnitude of the shift is larger than the random fluctuations of the sex-specific trait means ($|\Lambda_f|, |\Lambda_m| > \delta$), but smaller than, or on the order of, half the width of the fitness function ($|\Lambda_f|, |\Lambda_m| \lesssim 0.5\sqrt{V_S}$). The lower bound on shift sizes was motivated by a desire to consider only non-negligible shifts, and the upper bound was motivated by the fact that our analytic predictions for (asymptotic) phenotypic variation after the shift in optimum remain accurate in the range $\Lambda_f, \Lambda_m \lesssim 0.5\sqrt{V_S}$ (even for tests run in the extreme case of symmetric sex-specific selection and completely shared genetic architecture between the sexes; see Supplementary Section 3 and Supplementary Fig. 1).

We work in units of $\delta$, the typical deviation of the population mean from the optimum at equilibrium in the infinitesimal limit. Working in these units (by setting $V_S = 2N$ so that $\delta^2 = V_S/(2N) = 1$) makes our results invariant with respect to changing the population size, $N$, stabilizing selection parameter, $V_S$, mutational input per generation, $2NU$, and distributions of incoming effect magnitudes, $g(a)$.

## Simulations

For reasons of efficiency, our simulations are based on two additional simplifying assumptions. First, that alleles are at linkage equilibrium, allowing us to simulate the evolution of the population by tracking only the list of segregating alleles in the population, and their frequencies, rather than individuals. We refer to simulations in which we make this simplification as *Wright–Fisher* simulations because in each generation allele frequencies are updated according to a Wright–Fisher process. Second, we assume that allele frequency differences between sexes after selection are negligible (i.e. $x_f = x_m = x$ so alleles are at Hardy–Weinberg equilibrium). This assumption allows us to track only average frequencies of alleles, rather than sex-specific frequencies; and we refer to simulations which make this simplification as *Hardy–Weinberg* simulations. In Supplementary Section 4 and Supplementary Fig. 2, we provide more details about the assumptions behind each simulation type and test the robustness of our simulations to these two simplifying assumptions. We test the assumption of Hardy–Weinberg equilibrium by comparing the results of our *Wright–Fisher Hardy–Weinberg* simulations

**Table 1.** Summary of notation.

| Symbol | Definition |
| --- | --- |
| **General parameters** | |
| $N$ | Population size |
| $U$ | Expected number of mutations per generation per gamete |
| $L, \mu$ | The target size and the per site mutation rate (not specified with an infinite-sites model, where only the product $U \equiv L\mu$ is needed) |
| $V_S$ | Width of the Gaussian fitness function ($1/V_S \equiv$ strength of stabilizing selection) |
| $\delta$ | Typical magnitude of fluctuations around the optimum at equilibrium in the infinitesimal limit ($\delta^2 = V_S/(2N)$) |
| $\phi_a$ | Angle determining the fraction of stabilizing selection on an allele acting via each sex |
| $h(\phi_a)$ | Mutational distribution of $\phi_a$ |
| $h_r(\phi_a)$ | Simplified mutational distribution, with proportion $r$ of shared mutations and $1 - r$ of sex-specific mutations |
| $a^2$ | Squared overall phenotypic magnitude, corresponding to the scaled stabilizing selection coefficient ($a^2 \equiv 2Ns_e$ in units of $\delta^2$) |
| $g(a)$ | Mutational distribution of overall phenotypic magnitudes |
| $V_{A,O}$ | Overall additive genetic variance (defined in terms of the overall phenotypic magnitude) |
| $V_{A,*}$ | Additive genetic variance. For our choice $\gamma_f = \gamma_m = 1/\sqrt{2}$, $V_{A,O} = V_{A,f} = V_{A,m}$, which we call $V_{A,*}$, to indicate that * can be replaced with either of the two sexes |
| $V_{A,w}$ | Within-sex additive genetic variance, which corresponds to $V_{A,a}$ (appearing later in this table) |
| $V_{A,b}$ | Between-sex additive genetic variance |
| $V_{A,t}$ | Total additive genetic variance, computed across the two sexes as the sum of the within-sex plus between-sex variance, $V_{A,t} = V_{A,b} + V_{A,w}$ |
| $V_{A,e}$ | Additive genetic variance empirically calculated using the gene-expression dataset, averaged across sexes |
| **Sex-specific parameters** | |
| $\gamma_f, \gamma_m$ | Modulators of the relative strength of selection acting on females or males ($\gamma_f, \gamma_m > 0$ and $\gamma_f^2 + \gamma_m^2 = 1$) |
| $V_{S,f}, V_{S,f}$ | Widths of the sex-specific fitness functions, with $V_{S,f} \equiv 2V_S/(2\gamma_f^2)$ and $V_{S,m} \equiv 2V_S/(2\gamma_m^2)$ ($1/V_{S,f}$ and $1/V_{S,m}$ being the strengths of sex-specific stabilizing selection) |
| $a_f, a_m$ | Allele's sex-specific effects on the phenotype |
| $\bar{z}_f, \bar{z}_m$ | Sex-specific trait means |
| $SD_\pm$ | Signed sexual dimorphism, defined as $SD_\pm \equiv \bar{z}_f - \bar{z}_m$ |
| $SD$ | Sexual dimorphism, defined as $SD \equiv |SD_\pm|$ |
| $O_f, O_m$ | Sex-specific optima |
| $D_f, D_m$ | Sex-specific distances of the mean phenotypes from their respective optima |
| $\Lambda_f, \Lambda_m$ | Sex-specific shifts in trait optima |
| $V_{A,f}, V_{A,m}$ | Sex-specific additive genetic variances |
| $B$ | Between-sex covariance in the trait |
| $r_{fm}$ | Intersex correlation in the trait |
| $\mu_{3,f}, \mu_{3,m}$ | $\mu_{3,f} \equiv \frac{1}{2}(\mu_{3,fff} + \mu_{3,fmm})$ and $\mu_{3,m} \equiv \frac{1}{2}(\mu_{3,mmm} + \mu_{3,ffm})$, where $\mu_{3,\alpha\beta\gamma}$ ($\alpha, \beta, \gamma = f$ or $m$), are the third order central moments given by $\mu_{3,\alpha\beta\gamma} = \sum_i 2a_{i,\alpha}a_{i,\beta}a_{i,\gamma}x_i(1 - x_i)(1 - 2x_i)$ |
| $\tilde{F}_f, \tilde{F}_m$ | Female and male fixed backgrounds |
| **"Average" and "average distance" parameters**: $k_a \equiv \frac{1}{2}(k_f + k_m); \quad k_d \equiv \frac{1}{2}(k_f - k_m)$ | |
| $\bar{z}_a, \bar{z}_d$ | Average and average distance of the mean phenotypes |
| $O_a, O_d$ | Average and average distance of the phenotypic optima |
| $D_a, D_d$ | Distance between average and average distance of the mean phenotypes and their optima |
| $\Lambda_a, \Lambda_d$ | Shifts in average and average distance optima |
| $V_{A,a}, V_{A,d}$ | Average and average distance of the additive genetic variance |
| $\mu_{3,a}, \mu_{3,d}$ | $\mu_{3,a} = (\mu_{3,fff} + \mu_{3,fmm} + \mu_{3,ffm} + \mu_{3,mmm})/4; \quad \mu_{3,d} = (\mu_{3,fff} + \mu_{3,fmm} - \mu_{3,ffm} - \mu_{3,mmm})/4$ |
| $\tilde{F}_a, \tilde{F}_d$ | Average fixed background and fixed background difference |
| $F_a, F_d$ | Distance of the average and average distance fixed background from the optima ($F_a \equiv O_a - \tilde{F}_a$; $F_d \equiv O_d - \tilde{F}_d$) |

with *Wright–Fisher* simulations that track sex-specific allele frequencies; and we test the assumption of linkage equilibrium by comparing the results of *Wright–Fisher* simulations that track sex-specific allele frequencies with individual-based simulations.

Note that the robustness of our simulation results to these tests also provides justification for the fact that our analytic framework is robust, as it relies on the same two simplifying assumptions. In addition, the assumption of Hardy–Weinberg equilibrium is plausible a priori because we consider fairly weak selection. Previous studies have shown that sexually-antagonistic selection can lead to considerable differences in allele frequencies between the sexes, where balancing selection contributes to the maintenance of substantial genetic variation (Kidwell et al. 1977; Rice 1984; Morrow and Connallon 2013; Connallon and Clark 2014a). However, this requires very strong selection, beyond the range we consider in this study, and also beyond what is likely to apply to most traits.

In simulations, we let populations burn in for a period of 10N, 100N, or 500N generations (depending on the time each parameter combination takes to reach equilibrium, stated in the respective figure captions) to ensure they attain mutation–selection–drift balance, before applying the shift in optima or taking measurements when no shift in optima applies. In figures, we display averages and 95% CIs across replicates. Throughout, we simulate highly polygenic traits ($2NU \gg 1$) in two different parameter regimes, with genetic architectures that differ in such a way as to affect simulation results qualitatively. In the first parameter regime, simulation results are well-approximated by the infinitesimal model, which assumes that the trait is underlain by an infinite number of alleles, each with an infinitesimal effect size (Barton et al. 2017). For our modest shifts in optima, this will be the case when most mutations have fairly small effect sizes ($2Ns_e < 4$; corresponding to the *Lande case* in Hayward and Sella 2022). The second parameter regime, while still highly polygenic, has a significant contribution to trait variation from larger effect

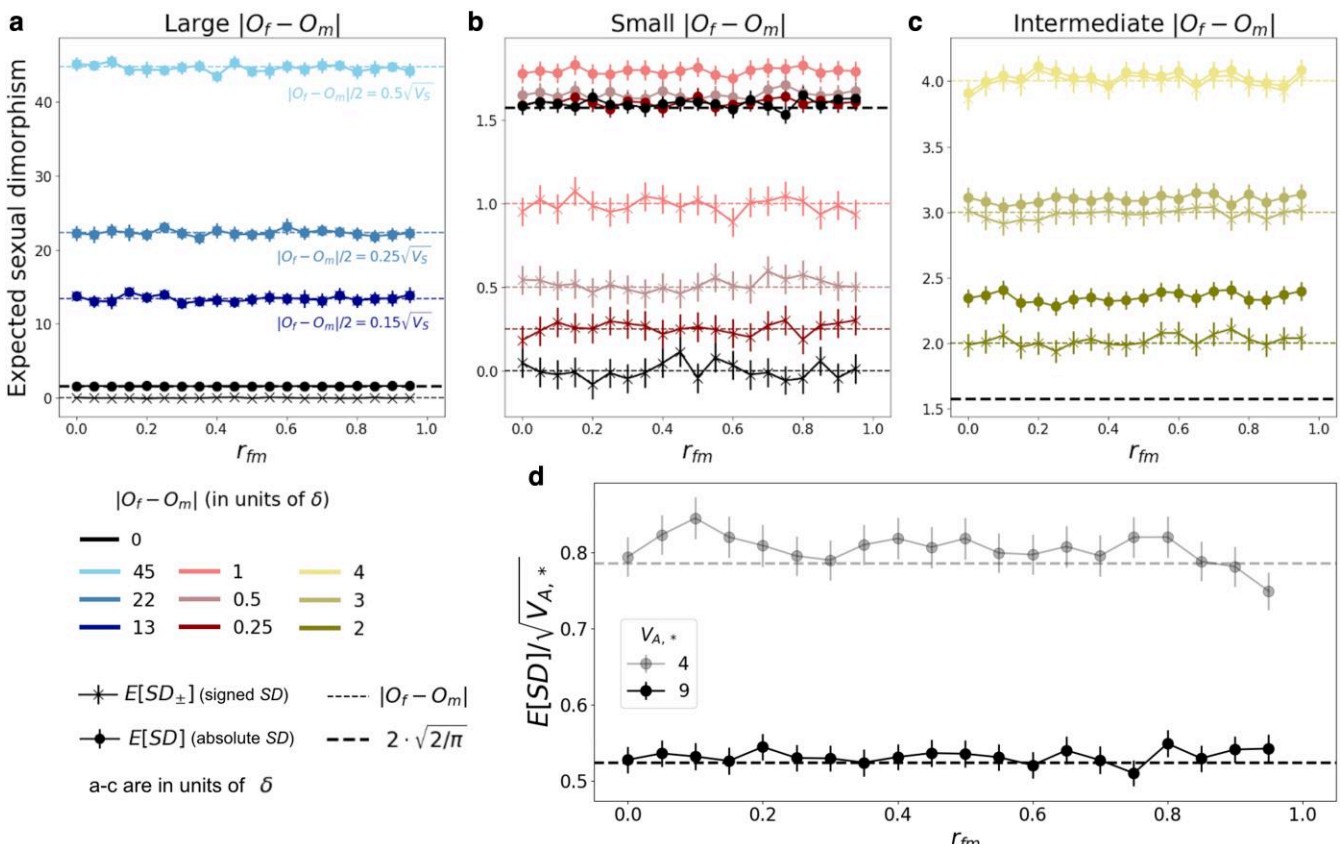

**Fig. 1.** Relationship between expected intersex correlation ($r_{fm}$) and sexual dimorphism at equilibrium with an approximately infinitesimal genetic architecture. a-c: Expected sexual dimorphism, signed (as the difference between sex-specific trait means, Equation (7); crosses) and absolute (as the absolute difference between sex-specific trait means, Equation (6); circles) across $r_{fm} \in [0, 1)$, with $V_{A,*} = 9$ and for various $|O_f - O_m|$ ranges (with the respective $|O_f - O_m|$ values indicated as dashed horizontal lines): a) large, with $|O_f - O_m| > 10$; b) small, with $|O_f - O_m| \in [0, 1]$; c) intermediate, with $|O_f - O_m| \in [2, 4]$. The thick black dashed line corresponds to $E[SD]$ predicted by Equation (33). d) Expected (absolute) sexual dimorphism, scaled by the standard deviation in sex-specific trait distributions, $E[SD]/\sqrt{V_{A,*}}$, for $O_f = O_m = 0$ and genetic variances $V_{A,*} = 4$ (semi-transparent) and 9 (opaque). Simulations with $V_{A,*} = 9$ were run for 100N generations, and simulations with $V_{A,*} = 4$ were run for 500N generations. Markers and error bars indicate estimates and 95% CIs calculated as 1.96·SEM across 2,000 replicates.

alleles (with $2Ns_e > 4$) and displays deviations from infinitesimal behavior when subject to directional selection (the *Non-Lande case* in Hayward and Sella 2022). We henceforth refer to these two types of genetic architecture as "approximately infinitesimal" and "multigenic," respectively.

To simulate traits with different degrees of intersex correlation, we relied on previous studies, which typically reduce the very complex regulatory genetic architecture of sex-specific trait expression into the consideration of shared and sex-specific mutations (Rhen 2000; Reeve and Fairbairn 2001; Bolnick and Doebeli 2003). In this case, we assume there is a proportion $r$ of shared mutations, with equal effect sizes in females and males ($a_f = a_m$), and the remaining $1 - r$ are sex-specific, out of which half are female-specific ($a_m = 0$) and half are male-specific ($a_f = 0$). For each mutation, there is an equal probability of it increasing or decreasing the trait value. This choice of trait architecture is extremely convenient because it gives us direct control over $r_{fm}$, as the expected intersex correlation exactly corresponds to the proportion of shared mutations ($E[r_{fm}] = r$; see the section on the intersex correlation at equilibrium for details). It is worth noting, however, that our analytic results do not rely on this simplification.

Here is a summary of the parameter values used in the simulations:

- In all simulations the population size is $N = 1,000$ and we take $\gamma_f^2 = \gamma_m^2 = 1/2$, so that the strength of stabilizing selection

is the same in both sexes and equal to the overall strength ($V_{S,f} = V_{S,m} = V_S$). In this case, sex-specific variances are equal and in referencing them in figures we replace the subscripts $f$ and $m$ with a general *, i.e. $V_{A,*} \equiv V_{A,f} = V_{A,m}$ (details in the section on the intersex correlation at equilibrium)

- In all simulations (except for Fig. 1), we consider an overall genetic variance of $V_{A,*} = 40$ (in units of $\delta^2$). With this choice, the width of the fitness function is about seven times larger that the standard deviation in the trait distribution (i.e. $\sqrt{V_S/V_{A,*}} \approx 7$), and the load due to additive phenotypic variance is about 1% (i.e. $1 - 1/\sqrt{1 + V_{A,*}/V_S} \approx 0.01$; Barton 1990).

- In order to illustrate the approximately infinitesimal and multigenic architectures, we consider different combinations of mutation rate $U$ and average squared effect size $E(a^2)$ (in units of $\delta^2$), sampled from an exponential distribution, yielding the same overall variance at equilibrium before the shift
  - Approximately infinitesimal architecture: $E(a^2) = 1$ (and $U = 0.0134$ for $V_{A,*} = 40$)
  - Multigenic architecture: $E(a^2) = 16$ (and $U = 0.0047$ for $V_{A,*} = 40$)

- We run simulations with various $E[r_{fm}]$ ($=r$) values, to illustrate the evolutionary outcomes with various genetic correlations between sexes. These correspond to choices of the

proportion of shared mutations of $r = 0.5$, $0.8$, and $0.95$ (except for Fig. 1, where we cover the whole $r_{fm}$ range).

- We typically implement shifts in sex-specific means of three concrete sizes. These correspond to $0.15\sqrt{V_S}$ (small), $0.25\sqrt{V_S}$ (medium), and $0.5\sqrt{V_S}$ (large). These magnitudes are within the limits of the shift size for our analytical approximations to work (tested in Supplementary Section 3). Relative to the equilibrium standard deviation of the phenotypic distribution (considering $V_{A,*} = 40$), the three shift sizes correspond to: $1.06\sqrt{V_{A,*}}$ (small), $1.77\sqrt{V_{A,*}}$ (medium), and $3.54\sqrt{V_{A,*}}$ (large).

- In Fig. 1, where we show simulation results for the dynamics at equilibrium, we explore a wide range of optimum differences ($|O_f - O_m|$). The large optimum differences correspond to our three shift sizes, the small optimum differences are less than or equal to $\delta = \sqrt{V_S/(2N)}$, and the intermediate optimum differences are between 2 and 4 ($\delta$).

Documented code for simulations can be found at https://github.com/gemmapuixeu/Puixeu_Hayward_2025.

## Results

In the present study, we examine the relationship between intersex correlation ($r_{fm}$) and sexual dimorphism ($SD$). The intersex correlation is defined as the ratio between the between-sex covariance, $B$ and the geometric mean of sex-specific variances, $V_{A,f}$ and $V_{A,m}$:

$$r_{fm} = \frac{B}{\sqrt{V_{A,f} V_{A,m}}}. \tag{3}$$

Under our assumptions of linkage equilibrium and an additive trait with no environmental contribution, $V_{A,f}$ and $V_{A,m}$ correspond to the sex-specific genic variances, which are the sum of the contributions to variance of all segregating alleles in each sex

$$V_{A,f} = \sum_i 2a_{i,f}^2 x_i(1 - x_i); \quad V_{A,m} = \sum_i 2a_{i,m}^2 x_i(1 - x_i) \tag{4}$$

where $x_i$ is the frequency and $a_{i,j}$ is the effect size of allele $i$ in sex $j$, for $j = f, m$. Similarly, under our assumptions, the intersex covariance, $B$, is given by the contributions to covariance of all segregating alleles

$$B = \sum_i 2a_{i,f} a_{i,m} x_i(1 - x_i). \tag{5}$$

It is important to note that such calculations for $r_{fm}$, $V_{A,f}$, $V_{A,m}$, and $B$ are only possible in simulations where sex-specific effects and allele frequencies are known. In empirical studies, other, "empirical" measures of sex-specific variances, intersex covariance, and intersex correlation are needed (see Supplementary Section 5 for more details).

The definition of sexual dimorphism is less universal than that of $r_{fm}$, as there are many ways to measure a dissimilarity between sex-specific trait means. In this study, we define sexual dimorphism to be the absolute value of the difference between sex-specific trait means

$$SD \equiv |\bar{z}_f - \bar{z}_m| \tag{6}$$

(where sex-specific trait means can be calculated by summing the allelic contributions to the mean $\bar{z}_f = \sum_i 2a_{i,f} x_i$ and $\bar{z}_m = \sum_i 2a_{i,m} x_i$). It is worth noting that some classical theoretical work (e.g. Lande

1980; Reeve and Fairbairn 2001) uses a signed difference in trait means to characterize sexual dimorphism

$$SD_{\pm} \equiv \bar{z}_f - \bar{z}_m \tag{7}$$

(actually, Lande 1980 and Reeve and Fairbairn 2001 consider $d \equiv \bar{z}_m - \bar{z}_f = -SD_{\pm}$ since they model sexual selection, in which the male optimum increases due to female mate preferences, but since sexes are interchangeable in our model this sign difference has no conceptual consequences). Nevertheless, most studies characterizing the relationship between intersex correlation and sexual dimorphism consider absolute measures. Most commonly, they consider the (sometimes error or average-normalized) absolute value of difference in trait means (McDaniel 2005; Ashman and Majetic 2006; Griffin et al. 2013) or absolute values of variations of the size dimorphism index (defined by Lovich and Gibbons 1992), obtained by subtracting one from the ratio of the trait mean of the larger sex to the trait mean of the smaller sex (Bonduriansky and Rowe 2005; Poissant et al. 2010; Leinonen et al. 2011). We choose to define $SD$ as the absolute value of the difference in sex-specific averages because, of the commonly used measures, it is simplest, and also the most similar to the signed characterization ($SD_{\pm}$) used in classical theoretical work—allowing us to make comparisons in a straightforward way. In addition, in order to easily evaluate the *significance* of deviations in $SD$ from zero, we sometimes scale it by the standard deviation of the phenotypic distribution.

We examine the relationship between intersex correlation and sexual dimorphism by characterizing the phenotypic and allele dynamics of a population at equilibrium under sex-specific stabilizing selection, mutation, and drift. In the section on $r_{fm}$ and SD at equilibrium, we describe the implications for the equilibrium relationship between intersex correlation and sexual dimorphism, extending classical work by considering the impact of drift. Then, in the section on exploring common hypotheses, we explore the conditions in which a correlation between $r_{fm}$ and $SD$ is expected by taking two common hypotheses typically invoked to explain a negative association in the literature as a starting point. Concretely, we explore the allelic and phenotypic response of a population (initially at equilibrium) to a change in sex-specific optima. We consider how these two common hypotheses are affected by assumptions made regarding (1) the genetic architecture of the trait (i.e. if the trait is approximately infinitesimal or multigenic) and (2) whether the two sexes evolve towards greater sex differences ($SD$ increases in response to a divergent shift; Box 1) or towards more similarity between the sexes ($SD$ decreases in response to a convergent shift).

Throughout our analysis, we rely on the fact that (under the continuous time approximation) allele dynamics, both in and out of equilibrium, can be described in terms of the first two moments of change in allele frequency in a single generation. The first moment of change, for an allele segregating at frequency $x$ with effect sizes $a_f$ and $a_m$ in females and males, respectively, is calculated by averaging the fitness of the three genotypes over genetic backgrounds, and is given by

$$E[\Delta x] = \underbrace{\left(\frac{a_f D_f \gamma_f^2}{V_S} + \frac{a_m D_m \gamma_m^2}{V_S}\right) x(1 - x)}_{\text{Directional selection}}$$
$$- \underbrace{\left(\frac{a_f^2 \gamma_f^2}{V_S} + \frac{a_m^2 \gamma_m^2}{V_S}\right)\left(\frac{1}{2} - x\right) x(1 - x)}_{\text{Stabilizing selection}}, \tag{8}$$

where $D_f \equiv O_f - \bar{z}_f$ and $D_m \equiv O_m - \bar{z}_m$ are the distances of sex-specific trait means from their respective optima (Equation (8) is derived in Section 1 of the Supplementary Material, and also in the supplementary material of Muralidhar and Coop 2024 with a different parameterization). The second moment is the standard drift term

$$V[\Delta x] \approx \frac{x(1-x)}{2N}. \tag{9}$$

The two terms in Equation (8) reflect two selection modes. The first corresponds to directional selection, which, within each sex, acts to increase (decrease) the frequency of those alleles which move sex-specific mean phenotypes closer to (further away from) sex-specific optima; its effect becomes weaker as the sex-specific distances to the optima, $D_f$, $D_m$, decrease. The second term corresponds to stabilizing selection, which acts to decrease alleles' contributions to phenotypic variance by reducing minor allele frequencies (MAFs); it weakens as the MAF approaches 1/2 (Simons et al. 2018). As a reminder, $V_{S,f} = V_S/(2\gamma_f^2)$ and $V_{S,m} = V_S/(2\gamma_m^2)$ correspond to sex-specific strengths of stabilizing selection, which we assume to be equal throughout. The relative importance of the two selection modes changes as $D_f$ and $D_m$ decrease, which allows us to define two phases in the allele dynamics (Jain and Stephan 2017; Hayward and Sella 2022): an initial, rapid phase, where directional selection acts to bring sex-specific means close to the new optima via allele frequency changes, and a later, equilibration phase, in which stabilizing selection drives alleles to loss/fixation at a slower pace. More details of these processes are provided when we examine the out-of-equilibrium dynamics in the section on exploring common hypotheses.

## Specifying the genetic architecture

Our choice to classify the genetic architecture as falling in one of two broad categories, *multigenic* and *approximately infinitesimal*, is in part motivated by equilibrium dynamics. At equilibrium, $D_f = D_m = 0$ in expectation and only the stabilizing selection term in Equation (8) is relevant

$$E_{eq}[\Delta x] = -\frac{a^2}{V_S}\left(\frac{1}{2} - x\right)x(1-x), \tag{10}$$

where we define $a > 0$ to be the *overall* phenotypic magnitude of an allele, with

$$a^2 \equiv a_f^2\gamma_f^2 + a_m^2\gamma_m^2. \tag{11}$$

Dynamics at equilibrium for a particular allele depend only on its equilibrium scaled selection coefficient which, it follows from Equation (10) (and given our choice to measure the trait in units of $\delta$, i.e. set $V_S = 2N$), equals its overall phenotypic magnitude:

$$2Ns_e \equiv 2Na^2/V_S = a^2/\delta^2 = a^2. \tag{12}$$

Consequently, allele frequency distributions at equilibrium depend only on the overall strength of selection on alleles (captured by $a^2$), allowing us to choose the equilibrium genetic architecture by specifying the distribution of allele magnitudes, $g(a)$. In the single-sex case, (provided the trait is highly polygenic) the degree of deviation from infinitesimal predictions following a shift in optimum can be precisely quantified, and depends largely on $g(a)$

(Hayward and Sella 2022). We find that this remains true with two sexes. When the distribution of incoming effect magnitudes is such that most incoming mutations have $a^2 = 2Ns_e$ on the order of 4 or smaller, approximations derived in the infinitesimal limit are highly accurate, and we therefore describe the genetic architecture as *approximately infinitesimal*. In contrast, when a significant fraction of incoming mutations have $a^2 = 2Ns_e > 4$, deviations from these approximations start to become appreciable, and we describe the genetic architecture as *multigenic*.

It is helpful to compare this approach to classical work in quantitative genetics, where less infinitesimal trait architectures are typically captured using a House of Cards model, which assumes that mutations replace the existing allelic effect at each locus with a new, randomly drawn value (Turelli 1984; Bürger et al. 1989; Zhang and Hill 2003). These models typically make three key assumptions. First, that a continuum of alleles is possible at each of a fixed number of loci (in contrast, we use a bi-allelic, infinite sites model). Second, that at each locus selection dominates mutation (by using an infinite sites model, we also implicitly make this second assumption). Third, that all alleles are subject to strong selection ($2Ns_e \gg 1$). It is important that we are able to relax this third assumption, both because distributions of new mutations completely lacking in nearly neutral or weakly selected alleles seem unlikely, and because weakly selected alleles play an important role in long-term dynamics following a shift in the optimum, even with a multigenic genetic architecture (Hayward and Sella 2022).

Although allele frequency distributions at equilibrium depend only on the overall strength of selection on alleles (captured by $a^2$), the intersex correlation depends on whether stabilizing selection is stronger when the allele is present in a female or when it is present in a male; which we parameterize in terms of an angle, $\phi_a$. This angle directly determines the fraction of stabilizing selection on an allele that acts via females ($\cos^2(\phi_a)$) and via males ($\sin^2(\phi_a)$) and corresponds to

$$\cos^2(\phi_a) = \frac{a_f^2\gamma_f^2}{a^2} \quad \text{and} \quad \sin^2(\phi_a) = \frac{a_m^2\gamma_m^2}{a^2} \tag{13}$$

(with $\cos(\phi_a)^2 + \sin(\phi_a)^2 = 1$). Parameterizing allele effects in terms of the allele magnitude $a$, and the angle, $\phi_a$ (rather than the sex specific effects $a_f$ and $a_m$), we can re-write the expected change in frequency at equilibrium under stabilizing selection (Equation (10)) as

$$E_{eq}[\Delta x] = -\underbrace{\frac{a^2}{V_S}}_{\substack{\text{total strength} \\ \text{of selection}}}\left[\overbrace{\underbrace{\cos^2(\phi_a)}_{\substack{\text{fraction selection} \\ \text{via females}}} + \underbrace{\sin^2(\phi_a)}_{\substack{\text{fraction selection} \\ \text{via males}}}}^{=1}\right]$$
$$\cdot \left(\frac{1}{2} - x\right)x(1-x). \tag{14}$$

We have chosen this parameterization because while the distribution of allele magnitudes, $g(a)$, directly determines whether the genetic architecture is approximately infinitesimal or multigenic and, as we will soon demonstrate, the distribution of angles, $h(\phi_a)$, determines the intersex correlation. However, (using $\gamma_f$ and $\gamma_m$) it is easy to recover the sex-specific effects from $a$ and $\phi_a$

$$a_f = \frac{a\cos(\phi_a)}{\gamma_f} \quad \text{and} \quad a_m = \frac{a\sin(\phi_a)}{\gamma_m}. \tag{15}$$

It should be noted that our analysis makes the assumption that $a$ and $\phi_a$ are independent, meaning that large-effect mutations are as likely to be female- or male-biased as small-effect mutations.

## The relationship between $r_{fm}$ and SD at equilibrium

We begin by recovering, in the context of a finite population, the classical result (previously derived in the deterministic limit of an infinite population size; Lande 1980; Wyman et al. 2013) that, at equilibrium, expected intersex correlation and *signed* sexual dimorphism (defined in Equations (3) and (7), respectively) are independent of each other (see the section on equilibrium E[SD$_\pm$] and $r_{fm}$). In a finite population, the relationship between absolute sexual dimorphism and signed sexual dimorphism is less straightforward; in the section on the effect of drift at equilibrium, we explore that relationship. We show that, although genetic drift generates deviations between $E[SD]$ and $|E[SD_\pm]|$ when sex specific optima are close, they are nevertheless both independent of the intersex correlation at equilibrium.

### Equilibrium E[SD$_\pm$] and $r_{fm}$ are independent

Under our assumption of an infinite sites model, and provided that at least some incoming mutations have different effects in the two sexes (i.e. $a_f \neq a_m$ for some alleles), directional selection will eventually drive the expected sex-specific means to their respective optima (Fig. 1a–c; Lande 1980; Wyman et al. 2013). Thus, at equilibrium

$$E[SD_\pm] = E[\bar{z}_f] - E[\bar{z}_m] = O_f - O_m. \tag{16}$$

Clearly, the expression for $E[SD_\pm]$ (signed sexual dimorphism) does not depend on intersex correlation. Also expected $SD$ (absolute sexual dimorphism) is independent from expected intersex correlation, because population dynamics are deterministic so

$$E[SD] = SD = |\bar{z}_f - \bar{z}_m| = |SD_\pm| = |E[SD_\pm]|. \tag{17}$$

To establish that expected equilibrium intersex correlation and expected signed sexual dimorphism are independent, it remains to derive an expression for expected $r_{fm}$ at equilibrium and show that it does not depend on trait optima or trait means. While this independence is already well established in the absence of genetic drift, we nevertheless include the results in the following section because our expressions—derived from a diffusion approximation—for equilibrium sex-specific variances, covariance, and the intersex correlation are novel. Readers less interested in mathematical results may prefer to skip directly to the section on the effect of drift at equilibrium.

*The intersex correlation at equilibrium.* In order to characterize the intersex correlation we need to calculate the second central moments of the phenotypic distribution ($V_{A,f}$, $V_{A,m}$, and $B$ defined in Equations (4) and (5)). To do so, it is useful to define an overall genetic variance which depends on alleles' overall phenotypic magnitudes (as defined in Equation (11))

$$V_{A,O} \equiv \sum_i 2a_i^2 x_i(1 - x_i). \tag{18}$$

Since Equation (10) for the expected change in frequency is identical to the single-sex case for an allele with magnitude $a$, the

overall variance is equal to the genic variance in the single-sex case and is given by

$$V_{A,O} = 2NU \cdot \int_0^\infty \upsilon(a)g(a)da, \tag{19}$$

where $g(a)$ is the distribution of incoming overall effect magnitudes and $\upsilon(a) = 4a \cdot D_+(a/2)$, where $D_+$ is the Dawson function Hayward and Sella (2022).

In Supplementary Section 2, we show that one can compute the expressions for sex-specific variances and covariance (relative to $V_{A,O}$) at equilibrium under stabilizing selection–mutation–drift balance as integrals over the distribution of angles, $h(\phi_a)$

$$\begin{aligned}
\frac{V_{A,f}}{V_{A,O}} &= \frac{1}{\gamma_f^2} \int_0^{2\pi} \cos(\phi_a)^2 h(\phi_a)\,d\phi_a; \\
\frac{V_{A,m}}{V_{A,O}} &= \frac{1}{\gamma_m^2} \int_0^{2\pi} \sin(\phi_a)^2 h(\phi_a)\,d\phi_a; \\
\frac{B}{V_{A,O}} &= \frac{1}{\gamma_f \gamma_m} \int_0^{2\pi} \cos(\phi_a)\sin(\phi_a)\,h(\phi_a)\,d\phi_a.
\end{aligned} \tag{20}$$

The expressions in Equation (20) can be combined to obtain the intersex correlation, yielding

$$r_{fm} = \frac{\int \cos(\phi_a)\sin(\phi_a)\,h(\phi_a)\,d\phi_a}{\sqrt{\int \cos(\phi_a)^2 h(\phi_a)\,d\phi_a \cdot \int \sin(\phi_a)^2 h(\phi_a)\,d\phi_a}}. \tag{21}$$

It is immediate from Equation (21), that the intersex correlation at equilibrium is independent of trait means and trait optima and therefore does not depend on the expected level of (signed) sexual dimorphism. In addition, Equation (21) shows that $r_{fm}$ at equilibrium depends only on the fraction of stabilizing selection acting on alleles via females (or males), which is determined by the distribution of angles $h(\phi_a)$.

As mentioned in the Simulations section, in simulations we use a specific, highly simplified distribution $h(\phi_a)$. In particular, we assume a proportion $r$ of mutations are shared, with equal effect sizes in the two sexes ($a_f = a_m$ and $\phi_a = \pi/4$ or $5\pi/4$), and a proportion $1 - r$ of mutations are sex-specific, out of which half are female-specific ($a_m = 0$ and $\phi_a = 0$ or $\pi$) and half are male-specific ($a_f = 0$ and $\phi_a = \pi/2$ or $3\pi/2$). For each mutation, there is an equal probability of its increasing the trait (i.e. $\phi_a = 0$, $\pi/4$, or $\pi/2$) or decreasing the trait (i.e. $\phi_a = \pi$, $5\pi/4$, or $3\pi/2$). Substituting this simplified distribution of angles into Equation (21) and performing the integrals (see Supplementary Section 2.2.1) yields $E[r_{fm}] = r$. This provides a simple way to control the expected $r_{fm}$: we choose $0 \leq r \leq 1$ and define $h_r(\phi_a)$ to be the simplified distribution described above with proportion $r$ of shared mutations. Note that, although we use this simplified distribution in simulations, our analytical results are derived for general distributions $h$, provided alleles are equally likely to be positive or negative (i.e, $h(\phi_a) = h(\phi_a + \pi)$, e.g. Equation (21)).

In simulations, in addition to using $h_r(\phi_a)$, we also typically assume that the overall strength of stabilizing selection is the same in both sexes ($\gamma_f = \gamma_m = 1/\sqrt{2}$). In this case, sex-specific variances are equal and in referencing them we can replace the subscripts $f$ and $m$ with a general $*$, i.e.

$$V_{A,*} \equiv V_{A,f} = V_{A,m} = V_{A,O}. \tag{22}$$

In addition, the intersex covariance is given by $B = rV_{A,*}$ and the variance from shared as well as sex-specific mutations equals to

$$V_{A,\text{shared}} = rV_{A,*} = B \tag{23}$$

$$V_{A,\text{sex-specific}} = (1 - r)V_{A,*}, \tag{24}$$

It is important to note that our expressions for $V_{A,O}$, $V_{A,f}$, $V_{A,m}$, $B$, $r_{fm}$, $V_{A,*}$, $V_{A,\text{shared}}$. and $V_{A,\text{sex-specific}}$ (Equations (19)–(24)) are actually expressions for the expected values of these quantities. Since, in this study, we consider only the expected values of the phenotypic variances, covariance, and correlations, we suppress the $E[\cdots]$ when referring to them, for ease of reading.

It is also worth noting that neither $V_{A,O}$ nor $V_{A,*}$ capture the *total* variance in the population, as would be empirically obtained across all the individuals of both sexes. This "total variance", $V_{A,t}$, can be computed from allele frequency data as the sum of the the within-sex and between-sex variance

$$V_{A,t} = V_{A,w} + V_{A,b}. \tag{25}$$

The concrete expressions for $V_{A,w}$, $V_{A,b}$, and $V_{A,t}$ can be found in Section 6 of the Supplementary Material.

Altogether, the results in this section show that, in expectation, between-sex correlation, $r_{fm}$, and signed sexual dimorphism, $SD_{\pm} = \bar{z}_f - \bar{z}_m$, are independent of each other at equilibrium. In particular, we see that (provided $r_{fm} < 1$) $E[SD_{\pm}] = O_f - O_m$ and that, consequently, when sex-specific optima coincide there will be no signed sexual dimorphism on average, irrespective of intersex correlation. While this is a well-established result in the literature (tracing back to Lande 1980), we additionally provide expressions to calculate the intersex correlation at equilibrium, showing that it depends on the distribution of angles $\phi_a$. Since

$$\tan(\phi_a) = \frac{a_m}{a_f} \cdot \frac{\gamma_f}{\gamma_m}, \tag{26}$$

it is apparent that the parameter $\phi_a$ depends both on the ratio of alleles' sex-specific mutational effects (i.e. $a_f/a_m$) and on the ratio of the strength of stabilizing selection in the two sexes (i.e. $\gamma_f/\gamma_m$). Thus, Equation (21) demonstrates that the presence of sex-specific variation (i.e. $r_{fm} < 1$) can arise from both sex-specific mutation ($a_f \neq a_m$) and sex-specific stabilizing selection ($\gamma_f \neq \gamma_m$), confirming the findings of other studies (e.g. Connallon and Clark 2014b).

### Drift generates nonzero E[SD] even when sex-specific optima coincide

In this section, we deepen our investigation of equilibrium dynamics by considering the impact of genetic drift. We show that, in finite populations, genetic drift can generate a nonzero average sexual dimorphism even when sex-specific optima are equal ($O_f = O_m$). However, the amount of dimorphism generated does *not* depend on the intersex correlation.

The nonzero dimorphism arises from the fact that—although, in expectation, at equilibrium trait means are equal to trait optima—genetic drift leads them to undergo rapid fluctuations around their expected values (Bürger and Lande 1994). This, in turn means that the difference in trait means, $SD_{\pm} = \bar{z}_f - \bar{z}_m$, will also typically undergo fluctuations. The only exception is when the intersex correlation is 1, with all incoming mutations having identical effect in both sexes ($a_f = a_m$). In this case, mean trait values in females and males must always coincide, and both signed sexual dimorphism and sexual dimorphism will be zero at all times (Supplementary Fig. 5; although $SD$ displays some increase due to new mutations, which arise sex-specifically, as discussed in Supplementary Section 7.2). Indeed, whenever the intersex correlation is high, short-term fluctuations in the two trait means are

highly correlated since most segregating variation has identical effects in both sexes (Supplementary Fig. 6c,d). However, provided $r_{fm} < 1$, mutations with effects that differ between the sexes will occasionally arise and fix, causing the two trait means to drift apart (over sufficiently long time periods). Consequently, at equilibrium sex-specific trait means will typically not be equal, $SD_{\pm} \neq 0$ (Supplementary Fig. 6), implying that $SD = |SD_{\pm}| > 0$ and hence that $E[SD] > 0$ (Fig. 1). It is easy to see that when trait values in the two sexes are uncorrelated ($r_{fm} = 0$), female and male trait means will fluctuate independently over both short and long time-scales (Supplementary Fig. 6a,b).

The fact that sexual dimorphism is nonzero for $r_{fm} < 1$ is a simple consequence of the fact that the variance in the difference in trait means is nonzero. Indeed, if the distribution of $SD_{\pm}$ were Gaussian, sexual dimorphism, and the variance in the difference in trait means would follow a very simple relationship:

$$E[SD] = E[|SD_{\pm}|] \approx \sqrt{\frac{2}{\pi}} \cdot \sqrt{V[SD_{\pm}]} = \sqrt{\frac{2}{\pi}} \cdot \sqrt{V[SD]}. \tag{27}$$

It turns out that the distribution of $SD_{\pm}$ is well-approximated by a Gaussian distribution for both an approximately infinitesimal and a multigenic genetic architecture (QQ plot in Supplementary Fig. 7), and Equation (27) performs remarkably well. Consequently, we can calculate expected sexual dimorphism by calculating its variance, which is more mathematically tractable.

We begin by finding an expression for $V[SD]$ in terms of the variance in population trait mean and the covariance in sex-specific trait means at equilibrium under stabilizing selection. From the definition of $SD$ it follows that:

$$\begin{aligned} V[SD] = V[SD_{\pm}] &= V[\bar{z}_f - \bar{z}_m] \\ &= V[\bar{z}_f] + V[\bar{z}_m] - 2Cov[\bar{z}_f, \bar{z}_m] \end{aligned} \tag{28}$$

We can re-write the expression above by considering the population mean phenotype, $\bar{z} = \frac{1}{2}(\bar{z}_f + \bar{z}_m)$. It has variance $V[\bar{z}] = \frac{1}{4}(V[\bar{z}_f] + V[\bar{z}_m]) + \frac{1}{2}Cov[\bar{z}_f, \bar{z}_m]$, implying that $V[\bar{z}_f] + V[\bar{z}_m] = 4V[\bar{z}] - 2Cov[\bar{z}_f, \bar{z}_m]$. Assuming that the magnitude of fluctuations in trait mean is equal between sexes, $V[\bar{z}_f] = V[\bar{z}_m]$, this gives us the size of sex-specific fluctuations around the optima

$$V[\bar{z}_{f/m}] = 2V[\bar{z}] - Cov[\bar{z}_f, \bar{z}_m]. \tag{29}$$

In Supplementary Fig. 8 we demonstrate, using simulation results for a wide range of $r_{fm} < 1$ and both an approximately infinitesimal and a multigenic genetic architecture, that $Cov[\bar{z}_f, \bar{z}_m] = 0$ (by showing that $1 - Cov[\bar{z}_f, \bar{z}_m]/V[\bar{z}] = 1$; Supplementary Fig. 8b). Putting this result together with Equations (28) and (29) reveals that both the magnitude of sex-specific fluctuations around the optima and variance in sexual dimorphism can be expressed in terms of the variance in population mean

$$V[\bar{z}_{f/m}] = 2V[\bar{z}] \text{ and} \tag{30}$$

$$V[SD] = V[\bar{z}_f] + V[\bar{z}_m] = 4V[\bar{z}]. \tag{31}$$

Fortunately, $V[\bar{z}]$ is theoretically predicted: the size of the fluctuations of the population mean around the optimum at equilibrium under stabilizing selection is $V[\bar{z}] = \delta^2 = V_S/(2N)$ Bürger and Lande (1994) which, for our choice of units, equals 1 (see also Supplementary Fig. 8a). Consequently, the magnitude of sex-specific fluctuations around the optima is given by $V[\bar{z}_{f/m}] = 2$ (Equation (30)). Some intuition for this result can be

gleaned by considering that it might arise from the fact that the population size of females and males is 0.5N, so that

$$V[\bar{z}_{f/m}] = \frac{V_{Sf/m}}{2N_{f/m}} = \frac{V_S}{2N/2} = 2\delta^2 = 2. \tag{32}$$

Also, following Equation (31), the expected variance in sexual dimorphism, $V[SD]$, when $O_f = O_m = 0$ is equal to 4, which we recover in Supplementary Fig. 8c. Finally, it follows from Equations (27) and (31) that when sex-specific optima coincide, the expected sexual dimorphism is given by

$$E[SD] = 2 \cdot \sqrt{\frac{2}{\pi}} \cdot \sqrt{V[\bar{z}]} = 2 \cdot \sqrt{\frac{2}{\pi}} \cdot \delta \approx 1.6. \tag{33}$$

This implies that, even when selection on the two sexes is identical, the typical value of $SD$ at equilibrium under stabilizing selection is nonzero and larger than the typical deviation of the population mean phenotype from the optimum, both with an approximately infinitesimal (Fig. 1b) as well as with a multigenic (Supplementary Fig. 3b) genetic architecture.

*Drift does not induce an association between $E[SD]$ and $r_{fm}$.* When sex-specific optima coincide, the prediction of $E[SD] \approx 1.6$ is supported by both theory (Equation (33)) and simulations for all values of $r_{fm} < 1$ (Fig. 1b; Supplementary Fig. 3b). It follows immediately that sexual dimorphism and intersex correlation are independent of each other when $O_f = O_m$. Simulation results reveal that this independence holds also for $O_f \neq O_m$. When the difference in sex-specific optima is nonzero and small ($|O_f - O_m| \lesssim 1$), the prediction of $E[SD] \approx 1.6$ for coinciding optima remains surprisingly accurate (Fig. 1b). This prediction holds across genetic variances ($V_{A,*}$) and for both approximately infinitesimal and multigenic genetic architectures (Supplementary Fig. 4b). For larger differences in optima ($|O_f - O_m| \gtrsim 4$), drift can be neglected and the absolute value of signed sexual dimorphism provides a good proxy for sexual dimorphism, i.e.

$$E[SD] \approx |E[SD_\pm]| = |O_f - O_m| \tag{34}$$

(Fig. 1a). For differences in optima between these two ranges, expected sexual dimorphism is greater than the absolute differences between sex-specific optima ($E[SD] \geq |O_f - O_m|$; Fig. 1c). Importantly, in all cases, expected sexual dimorphism and $r_{fm}$ remain independent of each other.

*The significance of drift-inflated $SD$.* As an interesting aside to exploring the relationship between sexual dimorphism and intersex correlation, we have established that even when trait optima coincide, genetic drift is likely to induce a nonzero sexual dimorphism. However, whether or not these deviations from zero in $E[SD]$ for $r_{fm} < 1$ are *of significance* depends on how their magnitude compares to the standard deviation in the sex-specific phenotypic distributions. Consequently, we can evaluate their significance by considering the unitless quantity $E[SD]/\sqrt{V_{A,*}}$. This scaling also provides a natural way to compare sexual dimorphism across different traits. Since $E[SD]$ is on the order of $\delta$, the effect of drift will be negligible for traits with genetic variance $\gg \delta^2$ (which is true for the regime we simulate in Figs. 2–4, which have $V_{A,*} = 40$; see the Simulations section). However, it may well be highly relevant for traits with genetic variance on the order of $\delta^2$ (or even $10\delta^2$).

In Fig. 1d, we show two such examples displaying $E[SD]/\sqrt{V_{A,*}}$ for traits with a fairly low genetic variance of $V_{A,*} = 4$ or 9, which correspond to fluctuations of the trait mean around the (shared) optimum with a typical magnitude of about half or a third of the

standard deviation in the trait distribution. For these particular (low phenotypic variance) examples, the effect of drift can be highly significant. Indeed, $E[SD]/\sqrt{V_{A,*}} = 0.8$ when $V_{A,*} = 4$ and $E[SD]/\sqrt{V_{A,*}} \approx 0.5$ when $V_{A,*} = 9$, implying that, just by chance, trait means in the two sexes could frequently differ by about a full or half of a phenotypic standard deviation, respectively. This is important because it suggests that special care should be taken before attributing even fairly large differences in female and male trait means to natural selection, especially for drift-sensitive traits such as gene expression (see the Discussion). The same results hold with a multigenic genetic architecture (Supplementary Fig. 3d).

It is worth noting that our model and most of our simulations assume linkage equilibrium which provides a good approximation for the dynamics with free recombination (see Supplementary Section 4). However, although a proper investigation into the effect of linkage disequilibrium is beyond the scope of this work, we speculate that more significant linkage disequilibrium might be expected to increase the importance of drift. This is because, in a finite population subject to stabilizing selection, linkage disequilibrium has the effect of decreasing the effective population size, and decreasing genetic variance in the trait (Santiago 1998). In addition, the decrease in effective population size might be expected to increase the size of the random fluctuations in the sex-specific optima (since genetic drift will be stronger).

## A negative relationship between $r_{fm}$ and $SD$—exploring common hypotheses

In the previous section, we describe how expected intersex correlation and sexual dimorphism are independent of each other at equilibrium. In this section, we explore the out-of-equilibrium dynamics of sex-specific adaptation in order to establish the conditions under which a relationship between intersex correlation and sexual dimorphism may emerge. Given that there is a widely held expectation of a negative correlation between the two (Bonduriansky and Rowe 2005; Fairbairn 2007; Griffin et al. 2013; Stewart and Rice 2018; McGlothlin et al. 2019), we interpret our results in light of the two hypotheses most commonly proposed to explain this expectation: first, that traits with ancestrally low $r_{fm}$ are less constrained to respond to sex-specific selection and therefore evolve to be more dimorphic (H$_1$); second, that sex-specific selection acts to reduce the intersex correlation (H$_2$).

We assess the applicability of these two hypotheses and the pattern they are expected to generate in the context of a population, initially at equilibrium under sex-specific stabilizing selection, mutation, and drift, that is subject to a sudden environmental change leading to a shift in sex-specific optima. In our analysis, we rely on the following equation describing how the per generation change in distances between sex-specific means and their optima ($D_f \equiv O_f - \bar{z}_f$ and $D_m \equiv O_m - \bar{z}_m$) depend on the second- and third-order central moments of the joint female and male phenotype distribution

$$
E\begin{bmatrix} \Delta D_f \\ \Delta D_m \end{bmatrix} = -\frac{V_S^{-1}}{2} \cdot \underbrace{\overbrace{\begin{bmatrix} 2\gamma_f^2 & 0 \\ 0 & 2\gamma_m^2 \end{bmatrix}}^{\gamma^2 \text{ matrix}} \overbrace{\begin{bmatrix} V_{A,f} & B \\ B & V_{A,m} \end{bmatrix}}^{G \text{ matrix}} \begin{bmatrix} D_f \\ D_m \end{bmatrix}}_{\text{Directional selection}}
$$
$$
+ \frac{V_S^{-1}}{2} \cdot \underbrace{\overbrace{\begin{bmatrix} 2\gamma_f^2 & 0 \\ 0 & 2\gamma_m^2 \end{bmatrix}}^{\gamma^2 \text{ matrix}} \overbrace{\begin{bmatrix} \mu_{3,f} \\ \mu_{3,m} \end{bmatrix}}^{\mu_3 \text{ matrix}}}_{\text{Stabilizing selection}}. \tag{35}
$$

Here $\mu_{3f} \equiv \frac{1}{2}(\mu_{3,fff} + \mu_{3,fmm})$ and $\mu_{3,m} \equiv \frac{1}{2}(\mu_{3,mmm} + \mu_{3,ffm})$, where $\mu_{3,\alpha\beta\gamma}$ ($\alpha, \beta, \gamma = f$ or $m$) equal $\mu_{3,\alpha\beta\gamma} = \sum_i 2a_{i,\alpha}a_{i,\beta}a_{i,\gamma}x_i(1-x_i)(1-2x_i)$ and

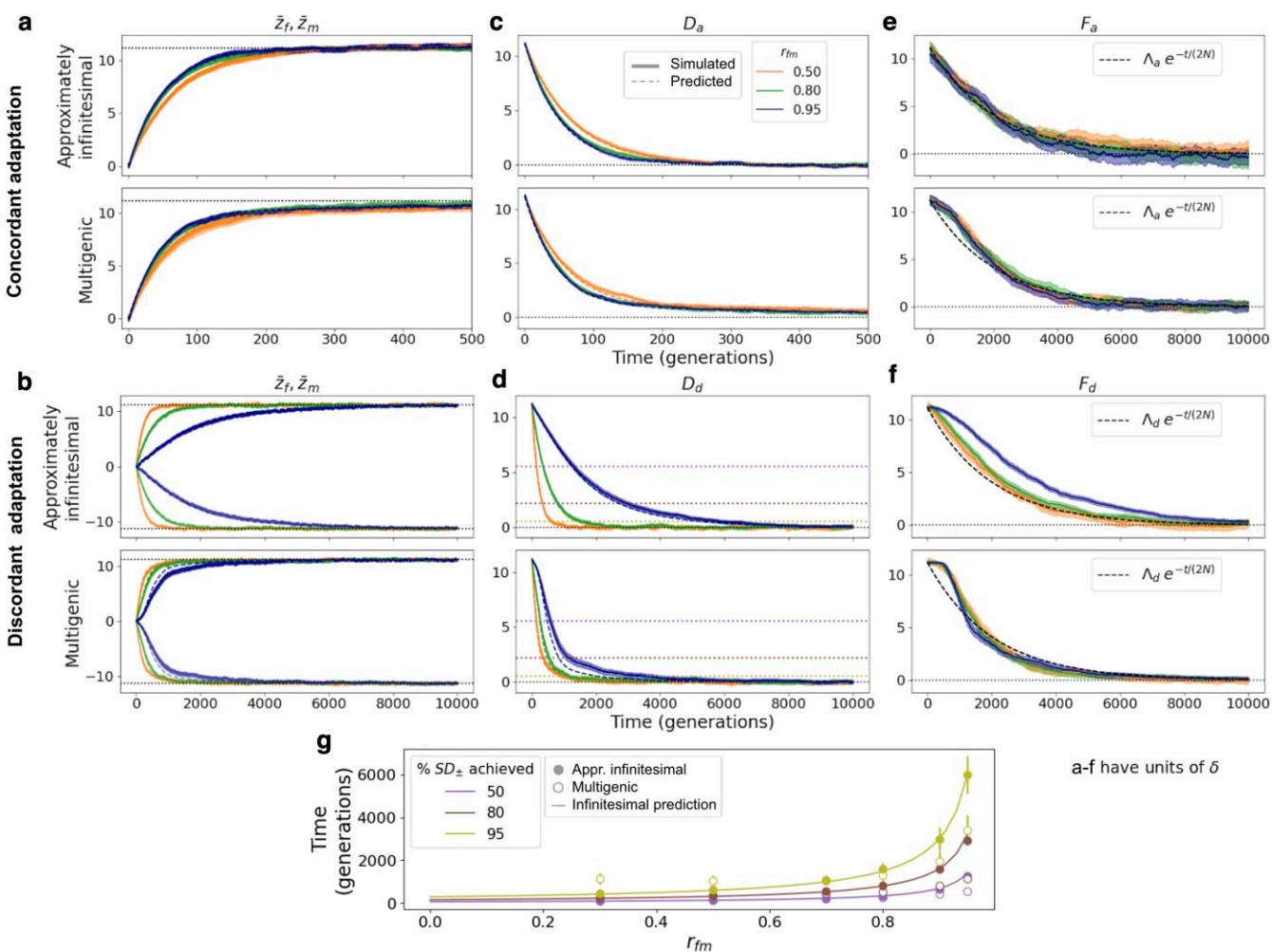

**Fig. 2.** Phenotypic evolution with an approximately infinitesimal ($E(a^2) = 1$, top panels) and multigenic ($E(a^2) = 16$, bottom panels) genetic architecture. a: Sex-specific trait means adapting to a shift in sex-specific optima of equal magnitude and direction, which implies only sexually-concordant adaptation ($\Lambda_a = 0.25\sqrt{V_S}$ and $\Lambda_d = 0$). b: Sex-specific trait means adapting to a shift in sex-specific optima of equal magnitude and opposite direction, which implies only sexually-discordant adaptation ($\Lambda_a = 0$ and $\Lambda_d = 0.25\sqrt{V_S}$). Sex-specific optima before the shift are both at zero, and after the shift are indicated as dotted lines. Thicker solid lines are simulations, and thin dashed lines are predictions using Equations (36) (approximately infinitesimal) and (46) (multigenic). c (d): $D_a$ ($D_d$) along time in simulations (thick solid lines) and predicted (thin dashed lines) using Equations (40) (approximately infinitesimal) and (47) (multigenic) for the sex-specific shifts in means in a (b). e (f): $F_a$ ($F_d$) along time for the optima shifts in a (b). Colored lines correspond to simulations and the dashed black line corresponds to the prediction according to Equations (49) and (50). g: Time to reach a given percentage of $SD_\pm$ (50, 80, and 90%, as purple, dark red, and olive circles; indicated as dotted horizontal lines in d) in simulations with approximately infinitesimal (solid) and multigenic (empty) genetic architectures and various levels of $r_{fm}$. Lines correspond to the infinitesimal prediction using Equation (43). All simulations have been run for 10N generations before the shift in optima, and for three levels of $r_{fm}$: 0.5 (orange), 0.8 (green), and 0.95 (blue; only g) has more $r_{fm}$ data points). Results display averages and 95% CIs computed as 1.96·SEM across 200 replicates. The x-axis in a and c spans a far shorter time period reflecting the fact that the initial phase of concordant adaptation tends to occur far more rapidly than discordant adaptation. All quantities displayed in a–f are in units of $\delta$.

are the third-order central moments of the joint female and male phenotype distribution. Equation (35) is derived by adding up the contributions to the change in mean phenotype coming from all segregating variants. Just like in the equation for alleles' expected change in frequency (Equation (8)), the two terms correspond to the two modes of selection underlying the dynamics: the first describes directional selection acting to reduce distances between means and respective optima at a rate that depends on sex-specific variances and covariance, while the second reflects the effect of stabilizing selection on an asymmetric (skewed) phenotypic distribution.

### Exploring $H_1$: $r_{fm}$ determines the rate of SD evolution

We begin by examining the relationship between intersex correlation and sexual dimorphism that might arise from the idea that traits with initially low intersex correlation respond

more rapidly to novel sex-specific selection, eventually achieving higher levels of sexual dimorphism—the first of the two hypotheses often invoked to explain why a negative correlation between $r_{fm}$ and SD is expected. As we saw in the section on equilibrium $E[SD_\pm]$ and $r_{fm}$ and in agreement with previous results assuming a polygenic or infinitesimal genetic architecture (Lande 1980), so long as there is variation for sexual dimorphism (i.e. if $r_{fm} < 1$), the two sexes will eventually evolve to diverge until sexual conflict is resolved—regardless of the intersex correlation (Fig. 1). However, while at equilibrium (signed and absolute) sexual dimorphism is independent of $r_{fm}$, the *rate* at which it evolves, and therefore the timescale for sexually-discordant evolution (i.e. evolution after a change in the distance between trait optima), is not. In this section, we characterize the time frame of adaptation to new sex-specific optima and its dependence on $r_{fm}$.

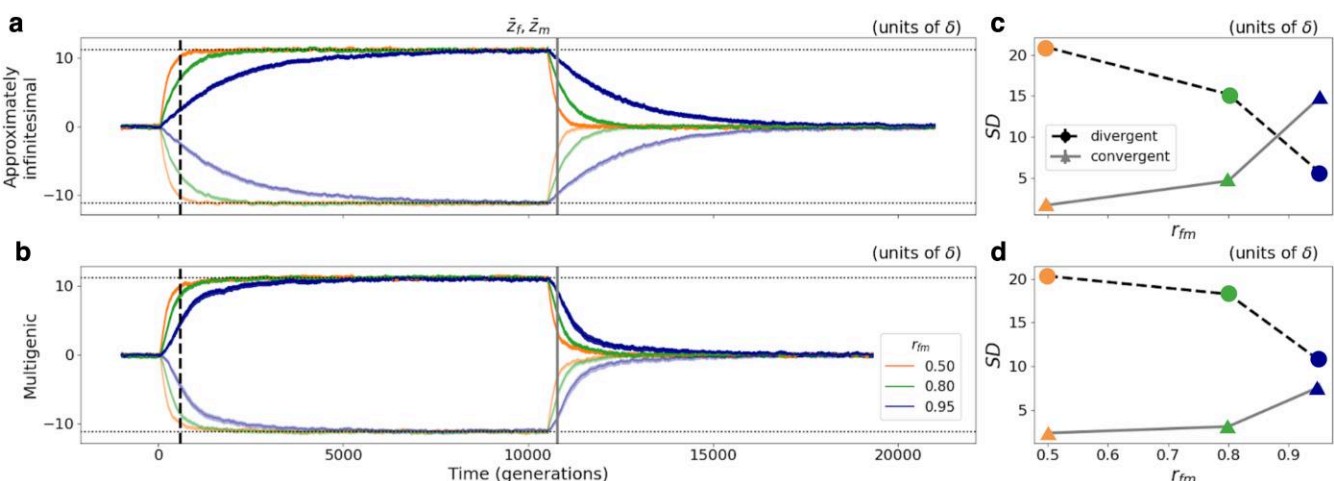

**Fig. 3.** Negative (positive) correlation between $r_{fm}$ and SD with divergent (convergent) adaptation. a (b): Sex-specific trait means adapting first to divergent and then to convergent shifts in optima of magnitude $0.25\sqrt{V_S}$ for approximately infinitesimal (multigenic) genetic architectures and three levels of $r_{fm}$: 0.5 (orange), 0.8 (green), and 0.95 (blue). c (d): Sexual dimorphism (given by the absolute value of the difference between sex-specific means, Equation (6)) for the three different levels of $r_{fm}$ at a given point of sexually-divergent—black dashed, corresponding to the timepoint of the black dashed vertical line in a (b)—and convergent—gray solid, corresponding to the timepoint of the gray solid vertical line in a (b)—adaptation, with an approximately infinitesimal (multigenic) genetic architecture. Results display averages and 95% CIs across 200 replicates.

As in the single-sex case, the timescale of sex-specific adaptation can roughly be split into two phases. An initial, rapid phase dominated by directional selection (first term in Equation (35)), where small changes in allele frequencies at many loci move the sex-specific means close to the new optima (which we refer to as the "rapid phase"); and a longer, stabilizing selection-dominated equilibration phase (second term in Equation (35)), during which the small frequency differences translate into a slight increase in the fixation probability of alleles with effects that align with the shifts in optima, relative to those with effects that oppose the shifts in optima (which we refer to as the "equilibration phase"). We examine the impact of intersex correlation on the time frame of both phases for sexually-concordant (i.e. the mean trait optimum across both sexes changes) or sexually-discordant (i.e. the distance between sex-specific optima changes; Box 1) adaptation of traits with approximately infinitesimal and multigenic architectures, and discuss the implications of our findings for the hypothesis $H_1$ that lower intersex correlation leads to increased sexual dimorphism. We find that because a high intersex correlation delays sexually-discordant evolution, intersex correlation might be correlated with the degree of sexual dimorphism at a given time during sex-specific adaptation. However, we also conclude that this correlation is only expected to be negative if selection typically favors increased dimorphism. If, on the contrary, selection more commonly favors decreased dimorphism, the association is expected to be *positive*.

### Adaptation in the infinitesimal limit: $r_{fm}$ determines the relative rate of sexually-concordant vs sexually-discordant evolution.

We first explore the rate of response to a change in sex-specific optima assuming an approximately infinitesimal genetic architecture. We also make the simplifying assumption that the strength of stabilizing selection is equal in the two sexes (i.e. $V_{S,f} = V_{S,m} = V_S$) so that the $\gamma^2$ matrix in Equation (35) is equal to the identity matrix. When the genetic architecture is approximately infinitesimal, phenotypic variances, and covariance remain almost unchanged after the shift in optima, and the trait distribution remains approximately symmetric ($\mu_{3,\alpha\beta\gamma} = 0$ for $\alpha, \beta, \gamma = f$ or $m$). Consequently, Equation (35) for the expected change in the distances of the sex-specific means from the optima reduces to

$$E\begin{bmatrix} \Delta D_f(t) \\ \Delta D_m(t) \end{bmatrix} = -\frac{V_S^{-1}}{2} \cdot \overbrace{\begin{bmatrix} V_{A,f}(0) & B(0) \\ B(0) & V_{A,m}(0) \end{bmatrix}}^{\text{G matrix}} \cdot \begin{bmatrix} D_f(t) \\ D_m(t) \end{bmatrix}, \quad (36)$$

which is the two-sex extension of the breeder's equation, as formulated by Lande (1980). Assuming that (co)variances remain constant along time ($V_{A,f}(0)$, $V_{A,m}(0)$, $B(0)$) this equation provides an accurate description of phenotypic evolution in the infinitesimal limit, where individual alleles do not change in frequency due to directional selection and the moments of the phenotypic distribution remain unchanged. From Equation (36), we see that after the shift in optima, directional selection acts directly on each sex to decrease the distance between the sex-specific trait mean and its optimum ($D_f(t)$ or $D_m(t)$) at a rate proportional to the distance itself, as well as to the initial phenotypic variance within that sex ($V_{A,f}(0)$ or $V_{A,m}(0)$). Directional selection within the opposite sex, however, can act to either increase or decrease the rate of adaptation to the new optimum at a rate proportional to the distance of the opposite sex from its new optimum, and to the intersex covariance, $B(0)$.

To better understand the role played by intersex covariance, we follow Lande (1980) (and others, e.g. Cheng and Houle 2020) in proposing a change of variables: instead of tracking sex-specific means ($\bar{z}_f$ and $\bar{z}_m$), we track the "average" and "average distance" of their means, given by

$$\bar{z}_a \equiv \frac{1}{2}(\bar{z}_f + \bar{z}_m) \quad \text{and} \quad \bar{z}_d \equiv \frac{1}{2}(\bar{z}_f - \bar{z}_m), \quad (37)$$

respectively. Notice that changes in $\bar{z}_a$ capture the evolution of the population as a whole (in fact, $\bar{z}_a$ is the population mean for the trait) and changes in $\bar{z}_d$ over time capture the evolution of signed sexual dimorphism, as

$$\bar{z}_d = 1/2 \cdot SD_\pm. \quad (38)$$

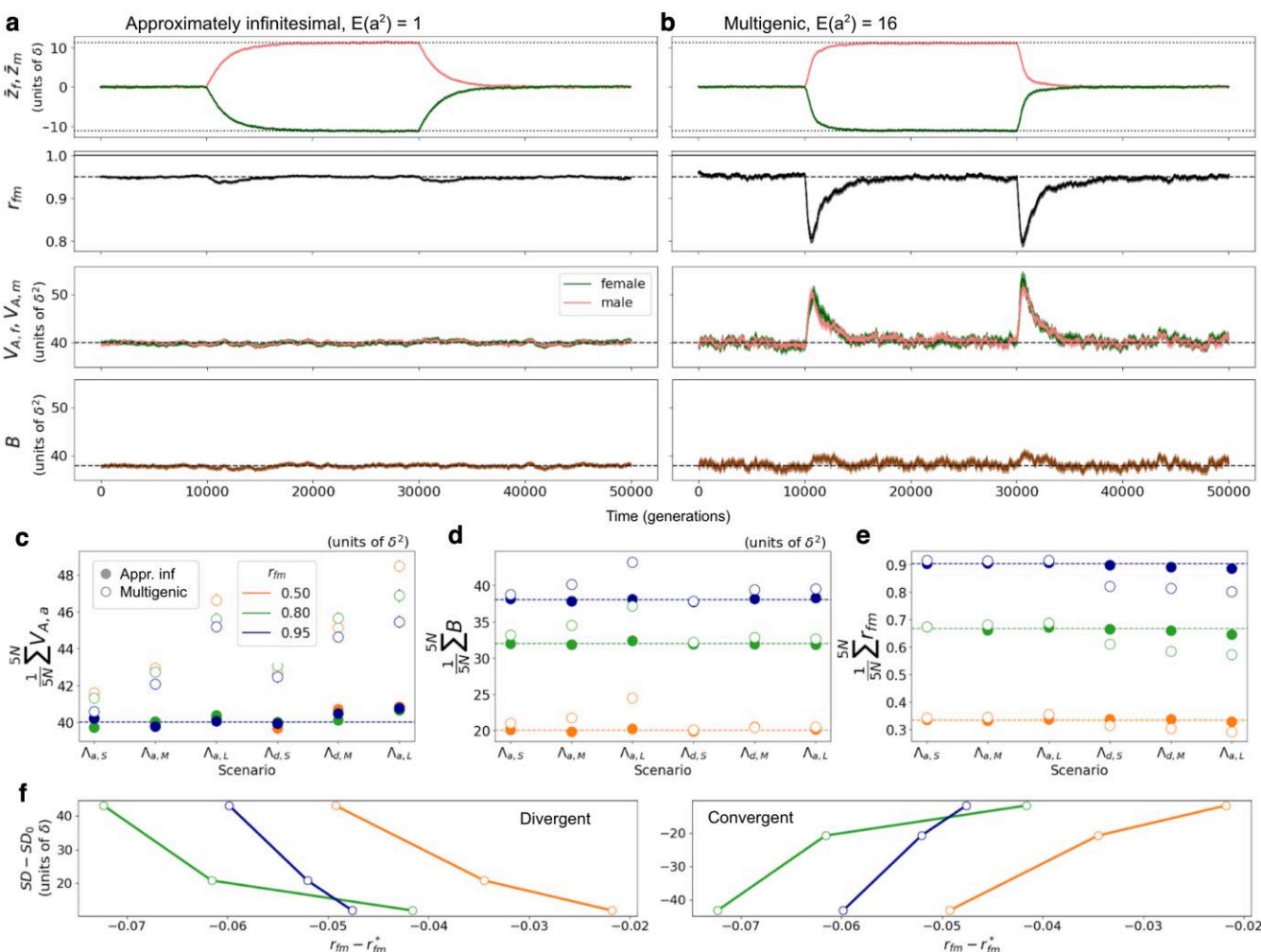

**Fig. 4.** Transient decrease in $r_{fm}$ during sexually-discordant (divergent and convergent) evolution. a,b: Evolution of sex-specific trait means ($\bar{z}_f$, $\bar{z}_m$), intersex correlation ($r_{fm}$), sex-specific variances ($V_{A,f}$, $V_{A,m}$), and covariance ($B$) along time with an approximately infinitesimal (a, $E(a^2) = 1$) and multigenic (b, $E(a^2) = 16$) genetic architecture, and with $r_{fm} = 0.95$. We let the population evolve for 10N generations before and after applying a shift in sex-specific optima of magnitude $\Lambda = 0.25\sqrt{V_S}$ inducing divergent (optima move apart), and then convergent (optima move together) evolution between the sexes. c, d, e: Means of average genetic variance ($V_{A,a}$; c), covariance ($B$; d), and intersex correlations ($r_{fm}$; e) across 5N generations after the shift in optima ($\equiv$ empirical integrals during the rapid phase of adaptation), for approximately infinitesimal (solid circles) and multigenic (open circles) genetic architecture and across different scenarios indicating different types of shifts: $\Lambda_{a,\_}$ are shifts of same magnitude and direction in both sexes, leading to sexually-concordant adaptation (similar to scenario depicted in Fig. 2a, in which $\Lambda_{d,\_} = 0$); $\Lambda_{d,\_}$ are shifts of same magnitude and opposite direction in both sexes, leading to sexually-discordant adaptation (similar to scenario in Fig. 2b, in which $\Lambda_{a,\_} = 0$). $\Lambda_{\_S}$, $\Lambda_{\_M}$, and $\Lambda_{\_L}$ indicate small, medium, and large shifts, with magnitudes $0.15\sqrt{V_S}$, $0.25\sqrt{V_S}$, and $0.5\sqrt{V_S}$, respectively. f: Negative (positive) relationship between intersex correlation and sexual dimorphism with divergent (left) and convergent (right) sexually-discordant selection. The y-axis corresponds to the difference between (theoretical predictions of) sexual dimorphism before and (long) after the shift, for the three shift magnitudes ($0.15\sqrt{V_S}$, $0.25\sqrt{V_S}$, and $0.5\sqrt{V_S}$); on the x-axis is the difference between the average $r_{fm}$ across 5N generations after the shift with a multigenic genetic architecture (corresponding to the open circles in $\Lambda_{d,S}$, $\Lambda_{d,M}$, and $\Lambda_{d,L}$ in e), and the equilibrium $r_{fm}$ values (dashed horizontal lines in e), for the three $r_{fm}$ (0.5, 0.8, and 0.95 in orange, green, and blue). a–e display averages and 95% CIs across 200 replicates.

Similarly, we define an "average" and "average distance" version of every variable $k$ that has both a female and male counterpart, as

$$k_a \equiv \frac{1}{2}(k_f + k_m); \quad k_d \equiv \frac{1}{2}(k_f - k_m). \quad (39)$$

So, for example, $O_a \equiv (O_f + O_m)/2$ and $O_d \equiv (O_f - O_m)/2$ are the average and average distance optima. With this change of variables, we can use Equation (36) to obtain an expression for the expected per generation change in $D_a \equiv O_a - \bar{z}_a$ and $D_d \equiv O_d - \bar{z}_d$:

$$E\begin{bmatrix} \Delta D_a(t) \\ \Delta D_d(t) \end{bmatrix} = -\frac{V_S^{-1}}{2} \cdot \overbrace{\begin{bmatrix} V_{A,a}(0) + B(0) & V_{A,d}(0) \\ V_{A,d}(0) & V_{A,a}(0) - B(0) \end{bmatrix}}^{G' \text{ matrix}} \cdot \begin{bmatrix} D_a(t) \\ D_d(t) \end{bmatrix} \quad (40)$$

From Equation (40) it follows that: (1) a high average phenotypic variance, $V_{A,a}(0)$, favors the evolution of both the overall trait mean (to the new mean optimum) and sexual dimorphism (to the new difference in optima); (2) a large, positive intersex covariance, $B(0)$, speeds up the evolution of the population mean to the new mean optimum, but delays the evolution of sexual dimorphism; (3) differences in phenotypic variance between the two sexes, $V_{A,d}(0) > 0$, generate interactions in the evolution of the overall trait mean, and sexual dimorphism.

If the initial phenotypic variance is the same in the two sexes, so that $V_{A,d}(0) = 0$, then the population mean and sexual dimorphism evolve independently and Equation (40) above reduces to

$$E[\Delta D_a(t)] = -\frac{(V_{A,a}(0) + B(0))}{2V_S} D_a(t);$$
$$E[\Delta D_d(t)] = -\frac{(V_{A,a}(0) - B(0))}{2V_S} D_d(t). \tag{41}$$

In continuous time this is solved by

$$D_a(t) = \Lambda_a e^{-t\frac{V_{A,a}(0)+B(0)}{2V_S}};$$
$$D_d(t) = \Lambda_d e^{-t\frac{V_{A,a}(0)-B(0)}{2V_S}}. \tag{42}$$

where $\Lambda_a$ and $\Lambda_d$ are the sizes of the shifts in $O_a$ and $O_d$.

Defining the length of the initial rapid phase of sexually-concordant ($t_a$) and sexually-discordant ($t_d$) adaptation to be the time that it takes for $D_a$ and $D_d$ to equal the typical deviation of the population mean from the optima at equilibrium, $\delta = \sqrt{V_S/2N}$, respectively, it follows that

$$t_a = \frac{2V_S}{V_{A,a}(0) + B(0)} \ \text{Ln}\left[\frac{\Lambda_a}{\delta}\right];$$
$$t_d = \frac{2V_S}{V_{A,a}(0) - B(0)} \ \text{Ln}\left[\frac{\Lambda_d}{\delta}\right]. \tag{43}$$

Thus the length of the initial phase of sexually-discordant adaptation relative to sexually-concordant adaptation is

$$\frac{t_d}{t_a} = \frac{V_{A,a}(0) + B(0)}{V_{A,a}(0) - B(0)} = \frac{1 + r_{fm}}{1 - r_{fm}}. \tag{44}$$

This result, initially obtained by Lande (1980), illustrates the quantitative constraint that intersex correlation places on the evolution of sex differences. In particular, when intersex correlation is close to 1, the denominator in Equation (44), $1 - r_{fm}$, will be very small, and sexually-discordant adaptation in the directional-selection dominated rapid phase could take orders of magnitude longer than sexually-concordant adaptation ($t_d \gg t_a$).

These dynamics are illustrated in the top panels of Fig. 2. Concretely, we implement sexually-concordant selection by applying sex-specific shifts in optima of the same magnitude and direction ($\Lambda_a > 0$, $\Lambda_d = 0$, shown in Fig. 2a), and sexually-discordant selection by applying sex-specific shifts in optima of the same magnitude but in opposite directions ($\Lambda_a = 0$, $\Lambda_d > 0$, shown in Fig. 2b), for low, intermediate, and high values of intersex correlation. We see that concordant adaptation happens at a much faster rate than discordant adaptation, and a higher $r_{fm}$ speeds up (slows down) concordant (discordant) adaptation, illustrated by a faster (slower) the reduction in $D_a$ ($D_d$) in Fig. 2c(d). This result holds qualitatively for both the approximately infinitesimal and the multigenic genetic architectures. However, the latter shows some important quantitative differences, as we outline in the next section.

Considering the simple relationship between $\bar{z}_d$ and $SD_{\pm}$ (Equation (38)), we obtain an expression for the signed sexual dimorphism over time

$$SD_{\pm}(t) = 2\Lambda_d \left(1 - e^{-t\frac{V_{A,a}(0)(1-r_{fm})}{2V_S}}\right) \tag{45}$$

Equation (45) shows that the amount of sexual dimorphism at a given time after a shift in sex-specific optima, depends on the shift in the difference between sex-specific optima, the strength of selection, the average genetic variance of the trait considered and the intersex correlation.

When differences in trait means are large or moderate ($|\bar{z}_f - \bar{z}_m| \gtrsim 4$; see the section on the effect of drift at equilibrium) we expect sexual dimorphism and signed sexual dimorphism to be similar. In order to explore out-of-equilibrium dynamics we follow previous theoretical work (Lande 1980; Reeve and Fairbairn 2001) and frequently consider $SD_{\pm}$ (or $\bar{z}_d = 1/2 \cdot SD_{\pm}$) as proxies for $SD$, especially when deriving analytic expressions. This removes the need to deal with (likely complicated) deviations between $SD_{\pm}$ and $SD$ introduced by genetic drift—deviations that are unlikely to be illuminating for our purpose of exploring the common hypotheses. Importantly, in figures explicitly intended to contextualize our results in terms of the two common hypotheses (Figs. 3c, d and 4f) we show results for the absolute $SD$ and these confirm intuitions gleaned from considering signed $SD$.

*Adaptation with a multigenic genetic architecture: Transient changes in the second- and third-order moments of the phenotype distribution alter the dynamics of phenotypic adaptation.* The accuracy of the predictions for the evolution of phenotypic means in Equations (36) and (40) relies on the assumption that the respective $G$ and $G'$ matrices remain constant over time. This will be approximately true when the genetic architecture is approximately infinitesimal. However, when considering a less infinitesimal trait architecture, with a significant proportion of mutations with larger effect sizes ($a^2 > 4$) as exemplified by our multigenic trait architecture, the approximations in Equations (36) and (40) are no longer accurate. This is because directional selection on alleles with larger effects can generate a significant increase in the second central moments of the joint phenotype distribution, as well as the establishment of nonzero third central moments (Supplementary Fig. 9). To accurately predict phenotypic evolution with a multigenic genetic architecture, we therefore need the full expression for the expected change in the distances of the sex-specific means from their respective optima (Equation (35)), with time-dependent second and third central moments (i.e. $V_{A,f}(t)$, $V_{A,m}(t)$, $B(t)$, $\mu_{3,f}(t)$, $\mu_{3,m}(t)$). Assuming, as we did for the approximately infinitesimal architecture, that the strength of stabilizing selection is equal in the two sexes (i.e. $V_{S,f} = V_{S,m} = V_S$) Equation (35) simplifies to

$$E\begin{bmatrix} \Delta D_f(t) \\ \Delta D_m(t) \end{bmatrix} = -\frac{V_S^{-1}}{2} \cdot \overbrace{\begin{bmatrix} V_{A,f}(t) & B(t) \\ B(t) & V_{A,m}(t) \end{bmatrix}}^{G \text{ matrix}} \cdot \begin{bmatrix} D_f(t) \\ D_m(t) \end{bmatrix}$$
$$+ \frac{V_S^{-1}}{2} \cdot \overbrace{\begin{bmatrix} \mu_{3,f}(t) \\ \mu_{3,m}(t) \end{bmatrix}}^{\mu_3 \text{ matrix}}. \tag{46}$$

As before, a simple change of variables (Equation (39)) yields an expression for the evolution of the overall trait mean (captured by $D_a$) and the level of sexual dimorphism (captured by $D_d$)

$$E\begin{bmatrix} \Delta D_a(t) \\ \Delta D_d(t) \end{bmatrix} = -\frac{V_S^{-1}}{2} \cdot \overbrace{\begin{bmatrix} V_{A,a}(t) + B(t) & V_{A,d}(t) \\ V_{A,d}(t) & V_{A,a}(t) - B(t) \end{bmatrix}}^{G' \text{ matrix}}$$
$$\cdot \begin{bmatrix} D_a(t) \\ D_d(t) \end{bmatrix} + \frac{V_S^{-1}}{2} \cdot \overbrace{\begin{bmatrix} \mu_{3,a}(t) \\ \mu_{3,d}(t) \end{bmatrix}}^{\mu_3 \text{ matrix}}. \tag{47}$$

By updating (co)variances and third central moments according to simulation results, we can use Equation (47) to accurately predict the mean trajectories of $D_a$ and $D_d$.

In cases where the trait has a multigenic genetic architecture, changes in the second and third central moments of the phenotypic distribution *do* affect the trajectories of mean phenotypes. However, they affect $D_a$ and $D_d$ in qualitatively different ways (see Supplementary Fig. 9 for changes in second and third central moments in both concordant and discordant adaptation, and Supplementary Fig. 10 for their separate effects on phenotypic evolution).

For concordant adaptation after a change in sex-specific means of equal magnitude and direction (captured by the decay of $D_a$), the dynamics are highly analogous to those observed in the single sex case Hayward and Sella (2022). Adaptation during the rapid phase occurs similarly as in the infinitesimal case and is well approximated by Equations (42)–(44) (Fig. 2a,c; Supplementary Fig. 10, top). However, after the rapid phase, the average trajectories of $D_a$ can deviate significantly from the exponential decrease predicted by Equation (42). Once the mean phenotype nears the new optimum, the system enters the equilibration phase when the decreasing distance and increasing third central moments reach the point at which the two terms on the right-hand side of Equation (47) approximately cancel out, and the changes in $D_a$ come almost to a stop (Fig. 2c, bottom, Supplementary Fig. 10, top). The rates of approaching the new optima are then largely determined by the rate at which the third central moments decay. This roughly corresponds to the rate at which the allele frequency distribution equilibrates (changes in frequency generated by directional selection translate into fixed differences, as we discuss in the section on high $r_{fm}$ delays equilibration). At this point, the system attains the original mutation–selection–drift balance around the new optima, and second and third central moments of the phenotypic distribution are restored to their equilibrium values. Indeed, the phenotypic dynamics for a trait with multigenic genetic architecture at the beginning of the equilibration phase are well captured by a quasi-static approximation (derived in Supplementary Section 8.1 and illustrated in Supplementary Fig. 11). We find that, while intersex correlation determines the time it takes to reach the equilibration phase (given approximately by Equation (43)), it does not seem to make a qualitative difference in the trajectories of the means during the initial part of the equilibration phase. However, as we demonstrate in the next section, a higher intersex correlation does imply a longer equilibration phase.

In contrast to concordant adaptation, discordant adaptation after a shift in sex-specific optima of equal magnitude and opposite directions (captured by decay in $D_d$), shows qualitatively different dynamics with a multigenic genetic architecture: concretely, changes in second and third central moments of the phenotypic distribution appear to respectively accelerate and decelerate phenotypic adaptation with respect to the infinitesimal predictions (Fig. 2b,d,g; Supplementary Fig. 10, bottom). Overall, we find that the time required for SD to evolve is substantially shorter under a multigenic architecture than under an infinitesimal one, especially for higher levels of $r_{fm}$ (Fig. 2d,g). For example, as Fig. 2g illustrates graphically, reaching 80% of the total difference in SD after a shift in optima (in dark red), it takes 50% and 220% longer with $r_{fm}$ of 0.8 and 0.95, respectively, under an infinitesimal genetic architecture compared to a multigenic one. These results suggest that the quantitative constraint that $r_{fm}$ poses on discordant adaptation derived in the infinitesimal limit (Equation (44)) is considerably relaxed when the genetic architecture deviates from this regime. This has potentially important implications for empirical studies, as the genetic architectures of traits in natural populations might well tend to be multigenic (see the *Discussion*).

*Higher intersex correlation delays equilibration for sex differences.* In the section on adaptation in the infinitesimal limit, we described how the time required for the average and average distance of the sex-specific trait means to approach their new optima depends on $r_{fm}$ (Equations (43) and (44)). These timepoints correspond to the length of the inital, directional selection-dominated phases of sexually-concordant and sexually-discordant adaptation, which are driven by small changes in allele frequencies at many loci. In this section, we analyze the timeframe associated with equilibration, during which stabilizing selection translates the allele frequency differences (generated by directional selection) between alleles with phenotypic effects that are aligned and opposed to the phenotypic shift into differences in fixation probabilities. This process restores the equilibrium phenotypic distributions with means at the new optima.

To examine the dynamics of equilibration we track the female and male fixed backgrounds ($\tilde{F}_f$ and $\tilde{F}_m$), defined as the trait values that females or males in the population would have if every segregating derived allele went extinct; and can be thought of as the component of the mean phenotypes maintained by fixations (as opposed to segregating variation). As before, we distinguish between sexually-concordant and sexually-discordant adaptation by performing a change of variables. Using Equation (39), we define the average fixed background and the fixed background difference ($\tilde{F}_a \equiv (\tilde{F}_f + \tilde{F}_m)/2$ and $\tilde{F}_d \equiv (\tilde{F}_f - \tilde{F}_m)/2$). Their distances from the new optima are

$$F_a = \Lambda_a - \tilde{F}_a; \quad F_d = \Lambda_d - \tilde{F}_d. \tag{48}$$

At equilibrium, we expect the fixed distances, $F_a$ and $F_d$, to be 0; the rate at which $F_a$ ($F_d$) approaches 0 gives the timescale over which sexually-concordant (sexually-discordant) equilibration occurs.

Not unexpectedly, we find that, in the approximately infinitesimal regime, sexually-concordant equilibration takes place at much the same rate as when there is just a single sex, and thus the trajectory of $F_a$ is well-approximated by

$$F_a(t) \approx \Lambda_a e^{-\frac{t}{2N}}. \tag{49}$$

(Hayward and Sella 2022; Fig. 2e). Sexually-concordant equilibration thus occurs over a time period on the order of 2N generations. Somewhat surprisingly, we find that when the intersex correlation is fairly low, $F_d$ also decays approximately exponentially at a rate 1/(2N)

$$F_d(t) \approx \Lambda_d e^{-\frac{t}{2N}}. \tag{50}$$

(Fig. 2f, top). When intersex correlation is high, however, the approximation in Equation (50) becomes quite inaccurate since the decay of $F_d$ can be quite delayed (Fig. 2f, top). Thus high intersex correlation increases the time period over which sexually-discordant equilibration occurs in the approximately infinitesimal case.

With a multigenic trait architecture, however, we observe slight deviations from exponential decay in $F_a$ and $F_d$ (even when intersex correlation is low). This is analogous to the deviations observed with a multigenic architecture in the single-sex case (Hayward and Sella 2022). In particular, the decay is initially slower and later faster then predicted by the approximations in Equations (49) and (50) (Fig. 2e,f, bottom). However, the time taken for the fixed backgrounds to reach the new optima, and therefore for the various moments of the phenotypic distribution to be restored to equilibrium values, is nevertheless on the order of 2N generations.

$r_{fm}$ *and* SD *might negatively or* positively *correlate.* We have shown that, while intersex correlation does not predict the overall realized sexual dimorphism, it does determine the rate at which it evolves. First, it directly determines the rate of sexually-concordant vs discordant phenotypic adaptation in the rapid phase; second, a high intersex correlation can delay sexually-discordant equilibrium. When considering non-equilibrium dynamics of adaptation, these aspects might contribute to generate an overall, negative relationship between $r_{fm}$ and sexual dimorphism, consistent with the first common hypothesis that initially lower intersex correlation allows for faster decoupling between sexes and more sexual dimorphism evolution. However, the same phenomenon can lead to a *positive* association between intersex correlation and sexual dimorphism in the case where selection acts to reduce sexual dimorphism.

This is because a lower intersex correlation allows for faster sexually-discordant evolution, both after a divergent as well as *convergent* shift in sex-specific optima (Fig. 3a,b). Concretely, after a divergent shift in sex-specific optima (i.e. increasing SD by keeping $O_a$ constant and increasing the absolute value of $O_d$, see Box 1), traits with a higher intersex correlation will take longer to diverge between sexes, leading to the commonly expected pattern of a negative relationship between intersex correlation and sex differences at a given time during divergent adaptation (black dashed line in Fig. 3c,d, corresponding to the timepoint of the black vertical dashed line in Fig. 3a,b). However, this is also true for adaptation after a convergent shift in optima (i.e. reducing SD by keeping $O_a$ constant and decreasing the absolute value of $O_d$): traits with a higher intersex correlation will take longer to adapt to a convergent shift than traits with an initially lower $r_{fm}$, potentially leading to the opposite pattern, i.e. to a *positive* relationship between intersex correlation and sex differences at a given time during convergent adaptation (gray solid line in Fig. 3c,d, corresponding to the timepoint of the gray vertical solid line in Fig. 3a,b). Importantly, because $r_{fm}$ imposes a stronger constraint on discordant adaptation in the infinitesimal case, this effect is markedly weaker when considering multigenic genetic architectures (Fig. 2b,d).

### Exploring H$_2$: *sex-specific directional selection* transiently reduces $r_{fm}$

In this section, we examine a second scenario with the potential to generate a correlation between $r_{fm}$ and SD. Specifically, we consider the idea that a correlation may emerge as a consequence of sex-specific adaptation driving a reduction in $r_{fm}$. This idea is prevalent in the literature and usually proposed as a hypothesis (often stated as an alternative to H$_1$) for why a *negative* correlation between the two might be expected. In order to examine its potential impact on the relationship between $r_{fm}$ and SD, we examine how $r_{fm}$ changes during sexually-discordant adaptation under a non-evolving genetic architecture.

Intersex correlation depends both on the variances within a single sex, $V_{A,f}$ and $V_{A,m}$, and on the covariance, $B$ (Equation (3)). In the section on adaptation in the infinitesimal limit, we established that for traits with approximately infinitesimal genetic architectures, the second-order central moments remain approximately unchanged by directional selection (Fig. 4a,c,d; see Supplementary Section 8 and Supplementary Fig. 9 for a more detailed discussion on the evolution of these moments). Consequently, when the trait has an approximately infinitesimal architecture, intersex correlation does not change at all (Fig. 4a, e). In contrast, as we discussed in the section on adaptation with a multigenic genetic architecture, for traits with multigenic architectures directional selection generates transient changes in second central moments of the phenotypic distributions (Supplementary Fig. 9 and Fig. 4b–d). These changes *can* result in a temporary decrease in intersex correlation [Fig. 4b,e; and also previously observed by Reeve and Fairbairn (2001)].

This decrease in intersex correlation (for traits with a multigenic architecture) is expected for sexually-discordant adaptation (i.e. when the distance between sex-specific trait optima changes), but not for sexually-concordant adaptation (i.e. when the mean optimum trait value changes for the two sexes equally; Reeve and Fairbairn 2001; Wyman et al. 2013). With sexually-concordant adaptation there is selection for phenotypic change along the main diagonal of the $G$ matrix (under our assumption that $V_{A,f} = V_{A,m}$), so there is an increase in sex-specific variance contributed by *shared* (but not sex-specific) mutations, which is equal to the increase in between-sex covariance (Supplementary Figs. 9b,c,e and 12). Consequently, intersex correlation, which is a ratio of the two, remains constant over time regardless of the magnitude of the shift (scenarios $\Lambda_{a,S}$, $\Lambda_{a,M}$, and $\Lambda_{a,L}$ in Fig. 4c–e). However, with sexually-discordant adaptation, directional selection drives an increase in frequency of those sex-specific mutations which drive phenotypic change in the direction of the shift, leading to an increase in sex-specific variances. Nevertheless, it does not on average increase the frequency of shared mutations, so covariance remains at equilibrium values (Supplementary Figs. 9h,i,k and 12), which leads to a decrease in $r_{fm}$ (scenarios $\Lambda_{d,S}$, $\Lambda_{d,M}$, and $\Lambda_{d,L}$ in Fig. 4c–e). This is only a transient phenomenon; as described in the section on high $r_{fm}$ delays equilibrium, (co)variances, as well as $r_{fm}$ are restored to their equilibrium values during the equilibration phase, over a time period on the order 2N (Fig. 2e,f).

The transient decrease in intersex correlation during sexual dimorphism evolution described above could generate an association between intersex correlation and sexual dimorphism, as suggested by the logic of H$_2$. However, the direction of this association depends on whether sexually-discordant adaptation is divergent (i.e. sex-specific optima move further apart) or convergent (i.e. sex-specific optima move closer together). For some intuition, let us consider a set of monomorphic (dimorphic) traits with similar $r_{fm}$ values at equilibrium, a subset of which becomes sex-specifically selected after a divergent (convergent) shift in sex-specific optima. Those traits in the process of diverging (converging) will experience a temporary decrease in intersex correlation, which would generate a negative (positive) correlation between $r_{fm}$ and sexual dimorphism. The negative (positive) association between intersex correlation and sexual dimorphism that might arise as a consequence of divergent (convergent) sexually-discordant adaptation is illustrated in Fig. 4f.

These results indicate that, in accordance with H$_2$, a negative correlation between intersex correlation and sexual dimorphism could arise from sex-specific adaptation leading to a reduction in $r_{fm}$. However, given our assumption of a non-evolving genetic architecture (see the *Discussion*), this phenomenon is transient. Moreover, it arises only under specific additional conditions. First, at least some traits must have a non-infinitesimal genetic architecture, where (co)variances change under directional selection. Second, traits must be adapting to (partially) discordant directional selection between sexes, where (a subset of) sex-specific mutations are more beneficial than shared mutations. Third, this sexually-discordant adaptation must be more often divergent than convergent. Notably, if adaptation is more frequently *convergent* than divergent, then the logic of H$_2$ would instead predict a

## Discussion

Based on the quantitative constraint that a high intersex correlation poses on the evolution of sexual dimorphism (Lande 1980, 1987; Stewart and Rice 2018) is the general idea that they should negatively correlate with one another. This idea arises either because it is predicted that traits will evolve to be more dimorphic if they are less correlated between the sexes (which we discuss as hypothesis $H_1$; Bolnick and Doebeli 2003; Poissant et al. 2010; Stewart and Rice 2018) or because there is an expectation that sexually-discordant evolution leads to a decrease in intersex correlation, which should allow independent adaptation of both sexes (which we discuss as hypothesis $H_2$; Lande 1980; Bonduriansky and Rowe 2005; Bonduriansky and Chenoweth 2009; McGlothlin et al. 2019).

Although these hypotheses are widespread in the sexual dimorphism literature—and supported in part by empirical findings of a generally (but not universally) negative correlation between $r_{fm}$ and sexual dimorphism across diverse taxa (Ashman 2003; Delph et al. 2004; Bonduriansky and Rowe 2005; McDaniel 2005; Poissant et al. 2010; Griffin et al. 2013)—they lack a clear mechanistic foundation. Specifically, it remains unclear under what conditions a correlation between $r_{fm}$ and SD should emerge, and what form that relationship should take. Addressing this gap is the central motivation of the present study: using models of sex-specific stabilizing selection, mutation, and drift, we investigate the conditions under which a correlation between intersex correlation and sexual dimorphism is expected, and identify the scenarios in which that correlation is negative or *positive*.

### At equilibrium $r_{fm}$ and SD are independent

First, we reproduce the well-known result (first obtained by Lande 1980) that, for a highly polygenic or quantitative trait with enough sex-specific genetic variation (either because there is enough standing variation or we have substantial sex-specific mutational input), sexual conflict will be resolved. That is, given enough time and as long as $r_{fm} < 1$, sex-specific means will eventually align with their optima (Fig. 1a–c). We illustrate that allele dynamics at equilibrium under stabilizing selection are independent of trait optima—and thus of trait means (Equation (10)); instead, they depend on the overall strength of stabilizing selection (Equation (12)). We show that the G matrix at equilibrium depends only on the overall and sex-specific mutational input and selection strength, which has also been shown for correlated traits in the 1-sex literature (Lande and Arnold 1983; Turelli 1985; Jones et al. 2003; Chantepie and Chevin 2020). This implies that, at equilibrium, the expected difference in trait means (signed sexual dimorphism) and expected intersex correlation are independent of each other (Fig. 1a–c).

### *Drift generates a nonzero* SD *even when sex-specific optima are aligned*

With a finite population, genetic drift generates random fluctuations in the sex-specific mean phenotypes. When sex-specific optima are far apart, these fluctuations can be neglected and sexual dimorphism is well-approximated by the difference in trait optima (for $r_{fm} < 1$) and, consequently, independent of intersex correlation. When trait optima coincide (or are close), however, random fluctuations can cause the expected *absolute value* of the difference in trait means (sexual dimorphism, SD) to differ

noticeably from their expected difference (signed sexual dimorphism, $SD_{\pm}$; Fig. 1a–c, Supplementary Fig. 3a–c). When intersex correlation is high, drift-induced fluctuations in SD are *slow* (on the time-scale of molecular evolution as they are generated largely by the rare fixation of mutations with sex-specific effects) and when intersex correlation is low they are rapid (generated largely by small fluctuations in frequency of standing variation with sex-specific effects). Nevertheless, we show that in both cases $E[SD] \approx 1.6 \cdot \delta$ where $\delta$ is the typical deviation of the population mean from the (shared) optimum. Since this result holds for all $r_{fm} < 1$, we find that, consistent with classical work, the magnitude of difference in trait means (sexual dimorphism) and intersex correlation are independent at equilibrium. Nevertheless, the result is of interest because it suggests that nonzero sexual dimorphism is actually *expected*—even in the absence of selection for such.

The significance of this (or any) nonzero sexual dimorphism depends on how it compares to the scale of genetic variation in the trait. Accordingly, whenever we make a point about the *magnitude* of sexual dimorphism, we scale SD by the standard deviation of the phenotypic distribution (Fig. 1d, Supplementary Fig. 3d). This standardization is often omitted in empirical work, where sexual dimorphism is computed based on averages and ignoring variances (e.g. Lovich and Gibbons 1992; Poissant et al. 2010). However, assessing the magnitude of sexual dimorphism—or comparing it across traits—is difficult without considering variation, as a given difference in sex-specific means is far more meaningful for traits with lower standard deviations. We therefore recommend that future studies report variance-standardized measures of sex differences.

With respect to drift-induced SD, we show that when fluctuations of the population mean are relatively large (of magnitude one-third or one-half of the genetic standard deviation of the trait distribution) then, just by chance, trait means in the two sexes could differ by almost a full and a half phenotypic standard deviation, respectively. The effect is expected to be smaller for traits with smaller fluctuation in means relative to phenotypic variance (i.e. higher $\sqrt{V_A}/\delta$). While empirical values of this ratio remain unknown, the effect of drift on SD at equilibrium may be particularly relevant for drift-sensitive traits with closely aligned sex-specific optima (Fig. 1b)—such as gene expression. We therefore urge caution in interpreting moderate or even fairly large sexual dimorphism in such traits as evidence of natural selection.

### A relationship between $r_{fm}$ and SD arises under sexually-discordant adaptation…

The two hypotheses most commonly invoked in the literature to explain a correlation between intersex correlation and sexual dimorphism—which is widely expected to be negative—both involve dynamic properties of the system. Using these hypotheses as a starting point, we explore the conditions in which a correlation between $r_{fm}$ and SD is expected to arise, and characterize its predicted patterns by analyzing the out-of-equilibrium dynamics of sex-specific adaptation under directional selection.

### … *because* $r_{fm}$ *constrains* SD *evolution (H$_1$)*

The first hypothesis, discussed in the section on exploring $H_1$, predicts higher levels of sexual dimorphism if intersex correlation is initially lower, leading to the expectation of a negative correlation between the two. We find that this can hold—but far from universally. Specifically, while intersex correlation does not determine the ultimate level of sexual dimorphism attained, it does determine the rate at which it evolves. As originally described by

Lande (1980), the rates of sexually-concordant and sexually-discordant evolution are proportional to $1 + r_{fm}$ and $1 - r_{fm}$, respectively (Equations (43) and (44), Fig. 2c,d), and therefore evolve on markedly different timescales when intersex correlation is high. This result illustrates the quantitative constraint that $r_{fm}$ imposes on the evolution of sex differences Lande (1980), and supports the idea that, after a limited time, the expected realized sexual dimorphism negatively correlates with intersex correlation (Bolnick and Doebeli 2003), providing apparent validation of this first hypothesis.

However, this result depends on two important considerations. First, the constraint that $r_{fm}$ poses on the evolution of SD is much weaker for a multigenic genetic architecture than the analytical results derived for the infinitesimal case suggest (Figs. 2d,g; and 3). Second—and more critically—as we show, high intersex correlation constrains not only *divergent* evolution between the sexes, but also *convergent* evolution. The latter has the potential to generate a *positive* relationship between intersex correlation and sexual dimorphism (Fig. 3). These considerations, discussed in more detail below, highlight the nuanced and non-universal relationship between $r_{fm}$ and SD.

In examining the timescales associated with sex-specific adaptation, we also obtain predictions for the timescale of equilibration. Hayward and Sella (2022) showed that for a single sex equilibrium is re-established over a time frame of the order of $2N$ generations. We find this result holds for sexually-concordant adaptation regardless of the $r_{fm}$ (Fig. 2e). However, when populations undergo sexually-discordant adaptation for traits with an approximately infinitesimal genetic architecture, we find that higher intersex correlation delays equilibration (Fig. 2f). Surprisingly, though, the effect of $r_{fm}$ on equilibration time is relatively modest, given the constraint it poses on the rate of phenotypic evolution. This suggests some compensatory process. With high $r_{fm}$, fewer mutations contribute to sexually-discordant adaptation compared to sexually-concordant adaptation, but those that do may experience stronger directional selection and thus fix more rapidly. As a result, the overall timescale of equilibration can remain similar across both modes of adaptation.

### …because the process of SD evolution involves a (transient) reduction in $r_{fm}$ (H₂)

The second hypothesis, typically proposed as an alternative to H₁ and discussed in the section on exploring H₂, supports the idea that sexually-discordant adaptation involves an accumulation of sex-specific mutations leading to a decrease in $r_{fm}$ over time. This idea traces back to Wright (1993) and Lande (1980, 1987), though neither author provides a mathematical justification. Rather, it appears to stem from an intuition of how such a process should evolve. Supporting this intuition, we find that (for a trait with a non-infinitesimal genetic architecture) intersex correlation decreases due to an increase in sex-specific variances, but not covariance, during sexually-discordant adaptation (Fig. 4).

These changes in the (co)variance matrix are *transient*; stabilizing selection translates the allele frequency changes between alleles with effects that are aligned and opposed to the phenotypic shift generated by directional selection into differences in fixation probabilities. In time, the transient increase in (co)variances ceases, and their equilibrium values are restored. Notably, the associated transient decrease in intersex correlation occurs during both divergent and convergent evolution (Fig. 4).

These dynamics suggest that sexual dimorphism can evolve without long-term changes in $r_{fm}$, as previously noted by Reeve and Fairbairn (2001) or Wyman et al. (2013). Reeve and

Fairbairn (2001)'s simulations further illustrate how this transient increase in second-order moments may both speed up sex-specific adaptation with a non-infinitesimal genetic architecture and lead to a transient decrease in $r_{fm}$—a pattern we also recover. However, that study (and, to our knowledge, others to date) does not discuss this phenomenon in the context of its potential contribution to generating a relationship between intersex correlation and sexual dimorphism—which is what we do here. We show that transient dynamics in the (co)variance structure can contribute to a correlation between $r_{fm}$ and SD, but that the direction of this relationship is not universally negative, contrary to prevailing intuition (see below for further discussion). Additionally, beyond the role of the (co)variances, we also examine the impact of third-order central moments in phenotypic adaptation. Specifically, we show that transient increases in third central moments can act to *slow* adaptation (Supplementary Fig. 11), consistent with findings from the single-sex case (Hayward and Sella 2022).

Our model predicts a transient reduction in $r_{fm}$ due to a temporary increase in sex-specific variances, rather than a lasting reduction due to decreased between-sex covariance—as is often suggested by verbal models dating back to Fisher (1958) and Lande (1980). This difference stems largely from the assumption of a fixed genetic architecture, an assumption shared by most models of sex-specific adaptation (Reeve and Fairbairn 2001; Bolnick and Doebeli 2003; Connallon and Clark 2014a, 2014b; Muralidhar and Coop 2024). In contrast, prevailing intuition appears to reflect scenarios in which the genetic architecture evolves over time, potentially reducing intersex covariance in a more lasting manner (Lande 1980; Wright 1993; Bonduriansky and Rowe 2005). Several biological mechanisms could facilitate such changes, e.g. sex-specific expression of autosomal loci, via sex-linked modifiers or alternative splicing mechanisms (McIntyre et al. 2006; Carreira et al. 2009; Stewart et al. 2010; Pennell and Morrow 2013; Singh and Agrawal 2023); and gene duplication followed by sex-specific regulation of the paralogs (Rice and Chippindale 2002; Proulx and Phillips 2006; Sison-Mangus et al. 2006; Connallon and Clark 2011), or sex-dependent dominance of alleles with sex-specific affects (Kidwell et al. 1977; Barson et al. 2015).

Mutations such as the acquisition of sex-specific regulatory elements (e.g. hormone sensitivity) or relocation of a gene to a sex chromosome could potentially increase the proportion of mutations with sex-specific effects on a trait under sex-specific stabilizing selection—assumed constant in our model. During the evolution of increased sexual dimorphism from an initially more monomorphic state, such mutations may be favored, as they accelerate the rate of divergence by decoupling the genotype-phenotype mapping between the sexes. Once fixed, these mutations result in a lasting reduction in $r_{fm}$ Wright (1993) and Bonduriansky and Rowe (2005), which in turn could stabilize a negative correlation between sexual dimorphism and intersex correlation (Williams and Carroll 2009; Stewart et al. 2010).

However, these changes are likely to occur slowly (Bonduriansky and Chenoweth 2009; Williams and Carroll 2009; Stewart et al. 2010), and may represent an additional phase beyond the two described by Lande (1980) and Lande (1987)—and reproduced here—for sexually-concordant and -discordant adaptation with a fixed genetic architecture, as also suggested by Wright (1993). Analyzing the dynamics with a constant genetic architecture is therefore a useful first step that likely reflects the most relevant genetic changes over the timescale of most experimental studies (e.g. Bird and Schaffer 1972; Reeve and Fairbairn 1996;

Stewart and Rice 2018). Indeed, our results show that transient changes in second-order central moments during directional selection on a polygenic trait (Reeve and Fairbairn 2001; Wyman et al. 2013) can generate a correlation between $r_{fm}$ and sexual dimorphism, even in the absence of any change in genetic architecture.

Allowing for an evolving genetic architecture—for example, by introducing modifier mutations that alter $h(\phi_a)$ and shift the proportion of sex-specific versus shared mutations—would be a natural extension of our model. Such modifications would allow us to ask whether faster evolution of sexual dimorphism promotes a more permanent reduction in $r_{fm}$, and whether these changes might be reversed once sexual conflict is resolved. Importantly, while the combination of sexually-discordant selection and an evolving genetic architecture may produce more persistent changes in $r_{fm}$ and potentially generate an association between intersex correlation and sexual dimorphism, our results predict that this association would not necessarily be negative—contrary to long-standing intuition. As we have demonstrated under a fixed genetic architecture, selection for *increased similarity* between the sexes from an initially more dimorphic state (i.e. convergent selection) can *also* favor reductions in $r_{fm}$. Exploring how intersex correlation and sexual dimorphism coevolve under a changing genetic architecture remains a promising direction for future work.

### The association between $r_{fm}$ and SD can be negative or positive

In contrast to prevailing intuition—which often assumes that the relationship between intersex correlation and sexual dimorphism is unequivocally negative—our results show that this is not necessarily the case. Instead, we find that the nature and strength of any such association depend on three key considerations.

First, a relationship between intersex correlation and sexual dimorphism is only expected to arise for traits that are out of equilibrium and undergoing (at least partially) sex-specific selection that alters the degree of sexual dimorphism—that is, during sexually-discordant adaptation. In contrast, when populations are at equilibrium, or when both sexes adapt together (i.e. under sexually-concordant adaptation), no relationship between $r_{fm}$ and SD is expected. Given the prevalence of sex-specific selection (Cox and Calsbeek 2009) and the often extended timescales required for the evolution of sexual dimorphism—especially for traits with high $r_{fm}$ and an approximately infinitesimal genetic architecture (Equations (43), Fig. 2d,g)—it seems likely that many traits are currently undergoing sexually-discordant directional selection, potentially generating an association between intersex correlation and sexual dimorphism.

Second, contrary to the widespread expectation that this correlation should be negative, our results show that its direction depends on the nature of the selection. Specifically, sexually-discordant that pulls the sexes farther apart (divergent evolution) is predicted to generate a negative correlation between $r_{fm}$ and SD, while selection that brings the sexes closer together (convergent evolution) is expected to produce a *positive* correlation (Figs. 3 and 4f). The fact that both $H_1$ and $H_2$ are typically interpreted as supporting a negative correlation suggests a general assumption in the literature that divergent evolution is more common than convergent evolution. However, empirical evidence shows that convergent evolution also occurs in some traits and species (Owens and Hartley 1998; Bond[uriansky 2006; Chursina 2019; Lassek and Gaulin 2022). In general, there is no compelling reason to assume that divergent evolution should dominate.

This suggests that the widely held intuition regarding a universally negative correlation between $r_{fm}$ and SD may reflect a bias in how sex-specific evolutionary processes are conceptualized.

Third, the genetic architecture of the trait has a major influence on the strength and character of the relationship between intersex correlation and sexual dimorphism. Interestingly, the two hypotheses have strongest effects with opposite genetic architectures. For $H_1$, although not strictly dependent on it, the effect is expected to be more significant for traits with an approximately infinitesimal architecture. This is because the constraint that $r_{fm}$ imposes on sexually-discordant adaptation—and thus its potential to generate a correlation—is more pronounced for traits with an approximately infinitesimal genetic architecture. This is consistent with insights from the literature on G-matrix evolution, which indicates that multigenic architectures are less constrained than approximately infinitesimal ones, as the genetic convariance structure is generally more stable under the latter (Lande 1979; Barton and Turelli 1987; Cai et al. 2024). This insight offers a complementary explanation for the disparities in the timescales of SD evolution observed across experimental designs. For example, our results suggest that a more rapid evolution of SD in some traits (as in e.g. Bird and Schaffer 1972) compared to others (as in e.g. Stewart and Rice 2018) may not be solely attributable to differences in $r_{fm}$ or sex-specific variances, as is often proposed. Rather, it may also reflect differences in genetic architecture: traits that evolve more rapidly may simply deviate more strongly from the infinitesimal regime—even if their $r_{fm}$ values are similar.

By contrast, we find that the transient reduction in $r_{fm}$ predicted under $H_2$ during sexually-discordant adaptation only occurs with non-infinitesimal genetic architecture. Under an infinitesimal model, the phenotypic distribution remains unchanged under directional selection, preventing such a reduction in intersex correlation. Empirical evidence from GWAS suggests that most complex traits do include large-effect mutations (e.g. Wood et al. 2014; Locke et al. 2015; Simons et al. 2018), indicating that many traits deviate from the infinitesimal regime and are therefore susceptible to experience the effect described in the context of $H_2$.

## Additional assumptions and limitations of our model

Our model relies on a number of assumptions, many of which we have already discussed—such as the use of a fixed genetic architecture. However, additional features of genetic architecture are also important to consider in shaping the dynamics of sex-specific adaptation. One such feature is the sex-specificity of individual mutations, which can significantly influence evolutionary outcomes (Rhen 2000). In this case, we draw overall squared effect sizes from an exponential distribution—a common choice in similar studies (e.g. Connallon and Clark 2014b)—and classify each mutation as either shared or sex-specific, with equal probabilities of being female- or male-specific. While this choice is common in many modeling frameworks (Rhen 2000; Reeve and Fairbairn 2001; Bolnick and Doebeli 2003), empirical evidence suggests the reality is more complex. Mutations often differ in both magnitude and direction of effect across sexes (Dimas et al. 2012; Oliva et al. 2020; Zhu et al. 2023; Puixeu et al., unpublished) and theoretical work has shown that such mutations may play a substantial role in sex-specific adaptation (Connallon and Clark 2014a; Muralidhar and Coop 2024). Although our theoretical results are compatible with these scenarios, explicitly incorporating such mutations into simulations would be a valuable extension of our

work. We also assume that the effect size distribution of new mutations is symmetric across sexes, and independent of the overall effect—that is, that strongly selected mutations are equally likely to be female- or male-biased. However, empirical evidence from *Drosophila* suggests a male bias in fitness effects of spontaneous mutations (Mallet et al. 2011; Sharp and Agrawal 2013), which may also warrant consideration in future models.

Importantly, previous work has shown that even with perfect intersex correlation and sexually-concordant selection, sexual dimorphism can still evolve if sex-specific genetic variances differ (Lynch and Walsh 1998; Connallon and Clark 2014b; Houle and Cheng 2021). This highlights that interpreting $r_{fm}$ as a constraint—as we and many others have done—relies on the assumptions that all genetic variance is additive, and that variances do not differ between the sexes (Lynch and Walsh 1998; Bonduriansky and Chenoweth 2009). More broadly, the interpretation of genetic correlations as the primary constraints on phenotypic adaptation has been criticized as overly simplistic and, in many respects, limiting for understanding the trade-offs and trajectories involved in evolutionary processes (Conner 2012; Cheng and Houle 2020; Houle and Cheng 2021). Together, these considerations underscore the importance of clearly stating the assumptions underlying any model, as different assumptions may lead to qualitatively distinct predictions. They also support the idea that differences in genetic architecture likely account for much of the variation in the evolutionary dynamics of sexual dimorphism that have been observed across species and traits.

In summary, our work provides an in-depth examination of the relationship between intersex correlation and sex differences as well as their joint evolutionary dynamics in a population adapting to a sex-specific shift in optima under sex-specific stabilizing selection, mutation, and drift—assuming a non-evolving genetic architecture. To our knowledge, it is the first comprehensive analysis to formalize and integrate multiple mechanisms that can generate an association between intersex correlation and sexual dimorphism, while also clarifying the assumptions that underlie these patterns. In doing so, it both synthesizes and challenges longstanding intuition in the field. More broadly, our findings emphasize the value of revisiting widely-used verbal arguments and demonstrate how placing them in an explicit theoretical framework can reveal hidden assumptions and yield deeper insights on the evolutionary forces shaping empirical patterns.

## Data availability

Documented code for simulations can be found at https://github.com/gemmapuixeu/Puixeu_Hayward_2025.

Supplemental material available at GENETICS online.

## Acknowledgments

We thank Tim Connallon for useful discussions and correspondence, Himani Sachdeva and Nick Barton for comments on the manuscript, and the Scientific Computing unit at ISTA for technical support.

## Funding

GP is the recipient of a DOC Fellowship of the Austrian Academy of Sciences at the Institute of Science and Technology Austria (DOC 25817) and received funding from the European Union's Horizon 2020 research and innovation program under the Marie Skłodowska-Curie Grant (agreement no. 665385). LKH received funding from the European Research Council, under the HaplotypeStructure Grant (Grant no. 101055327) to Nick Barton. The funders did not play any role in the study design, data collection and analysis, decision to publish, or preparation of the manuscript.

## Conflict of interest

None declared.

## Author contributions

GP and LKH conceived the study, performed the analyses, and wrote the manuscript.

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

*Editor: G. Coop*