## [Peer Review File · Genetics]

The relationship between sexual dimorphism and intersex correlation: do models support intuition?

Gemma Puixeu and Laura Hayward

NOTE: The reviews and decision letters are unedited and appear as submitted by the reviewers.

In extremely rare instances and as determined by a Senior Editor or the EIC, portions of a review may be redacted. If a review is signed, the reviewer has agreed to no longer remain anonymous.

The review history appears in chronological order.

Review Timeline:

Submission Date:	2025-01-30
Editorial Decision:	2025-03-21
Resubmission Received:	2025-06-09
Editorial Decision:	2025-07-21
Revision Received:	2025-07-25
Accepted:	2025-08-07

March 20, 2025

GENETICS-2025-307834

The relationship between sexual dimorphism and intersex correlation: do models support intuition?

Dear Dr. Hayward:

Two experts in the field have reviewed your manuscript, and I have read it as well. While your manuscript is not currently acceptable for publication in GENETICS, we would welcome a substantially revised manuscript. Both reviewers have comments and concerns to be addressed in a revised manuscript. You can read their reviews at the end of this email.

Both reviewers have concerns that should be addressed. Reviewer 2 in particular raises important concerns about the extent to which the authors test the hypotheses they lay out as their motivation and the extent to which previous papers have already addressed these concerns. We look forward to receiving your revised manuscript. Please let the editorial office know approximately how long you expect to need for revisions.

Upon resubmission, please include:

1. A clean version of your manuscript;
2. A marked version of your manuscript in which you highlight significant revisions carried out in response to the major points raised by the editor/reviewers (track changes is acceptable if preferred);
3. A detailed response to the editor's/reviewers' feedback and to the concerns listed above. Please reference line numbers in this response to aid the editor and reviewers.

Your paper will likely be sent back out for review.

Additionally, please ensure that your resubmission is formatted for GENETICS
<https://academic.oup.com/genetics/pages/general-instructions>

Follow this link to submit the revised manuscript: Link Not Available

Sincerely,

Graham Coop
Associate Editor
GENETICS

Approved by:
Sarah Otto
Senior Editor
GENETICS

Reviewer #1 :

This manuscript explores the theoretical relationship between sexual dimorphism (SD) and intersex genetic correlation under models of sex-specific stabilizing selection, mutation, and genetic drift. The authors critically examine the intuitive expectation of a negative correlation between SD and intersex genetic correlation by analyzing equilibrium and transient evolutionary dynamics across different genetic architectures (infinitesimal vs. multigenic).

Overall, the results are well-structured, comprehensive, and thoroughly discussed. Two particularly interesting findings are:

1. Convergent sexually dimorphic adaptation may lead to a positive association between intersex correlation and SD.
2. Genetic drift alone can generate non-zero absolute SD.

With these results, I have two key thoughts:

1.) I'm just guessing - a strong intersex genetic correlation may buffer the effect of genetic drift, potentially reducing SD when sex-specific optima are similar. This could contribute to the commonly observed negative relationship between intersex correlation and SD. I would be interested in further analysis of this effect-specifically, how intersex correlation influences non-

zero SD generated by drift-as it may provide a mechanistic explanation for the negative association. However, I leave this to the editors' and authors' discretion.

2.) Under fluctuating selection around a mean optimum, non-zero SD may emerge when the mean optima for both sexes coincide. In such a scenario, I hypothesize that intersex correlation would negatively correlate with SD. This could be the basis for a separate study. Personally, I find the interplay between genetic drift, intersex correlation, and fluctuating selection particularly intriguing and worth exploring in future work to better explain the relationship between rfm and SD.

Specific Comments:

Lines 744/ 813: The assumption that multigenic architectures are less constrained than the infinitesimal model is supported by literature on G-matrix evolution. The genetic covariance structure is generally more stable under the infinitesimal model (e.g., Barton & Turelli, 1987, 1989; Lande, 1979) than under a more Mendelian-like architecture, resulting in stronger constraints under the infinitesimal model. For further discussion and related empirical evidence, see the discussion section of Cai et al. Dissecting genetic correlation and pleiotropy through a genetic cross.

Genetic Architecture: Why were these two genetic architectures (infinitesimal vs. multigenic) chosen for comparison? How does the multigenic model relate to the classical "House of Cards" (HoC) model? Are they conceptually similar? It would be beneficial to mention the HoC model in the introduction to clarify its relationship with the frameworks used in this study.

Line 947 repeated words: 'similar'

There is no code in the provided github site.

Cheers,
Haoran Cai

Reviewer #2 :

This paper examines whether the evolution of sexual dimorphism in a quantitative trait is associated with a negative intersex genetic correlation in that trait. Specifically, this paper analyzes models of a polygenic trait, along with simulations of those models, to test different hypotheses for why there should be a negative association between the intersex genetic correlation and the level of phenotypic divergence between males and females. While there are interesting results in this paper, I have major concerns about whether the authors are actually able to test their hypotheses for the association between sexual dimorphism and the intersex genetic correlation. These, and other comments, are detailed below.

--- The authors state that they will test two hypotheses that can explain negative correlation between the level of sexual dimorphism and the intersex genetic correlation. The first posits that a high intersex correlation means that the rate of adaptation in a sexually dimorphic direction (relative to a sexually concordant direction) is slower, although, given an imperfect genetic correlation, the different male and female optima will eventually be reached. This is, as the authors note, originally stated in Lande 1980, and very well established in the quantitative genetic literature (e.g. Poissant et al. 2010, Wyman et al. 2013) in the context of sexual dimorphism, and more broadly, in the context of constraints on multivariate adaptation (Walsh & Blows 2009). The authors confirm that the rate of adaptation in a sexually dimorphic direction is constrained by the intersex correlation, and therefore confirm hypothesis 1.

The second hypothesis is that the genetic basis of the intersex genetic correlation will itself evolve in response to repeated bouts of discordant selection in males and females. In quantitative genetic terms, the genetic variance-covariance matrix, or the underlying mutational structure giving rise to that matrix, can evolve such that the intersex genetic correlation is reduced (a similar logic has been applied to explain how fluctuating selection can drive the evolution of a modular genetic architecture, see, e.g., de O & Whitlock 2023). But the authors do not test this hypothesis, because they assume that the genetic architecture of the trait stays constant. As would be expected under this regime, and as noted in prior papers (Reeve and Fairbairn 2001, Wyman et al. 2013), there are only transient changes in the intersex correlation under these assumptions, because the intersex genetic correlation is essentially prevented from evolving.

Given the previous work establishing hypothesis 1 in very similar regimes to those the authors consider, and the fact that the authors can only test a very limited version of hypothesis 2 due to their modelling assumptions, it is difficult to see why the paper is framed in terms of testing these two hypotheses at all. I think the results of the authors detailing the evolutionary dynamics of sexual dimorphism and the intersex genetic correlation under a multigenic, rather than infinitesimal, genetic architecture, could present an interesting twist on the existing literature on the evolution of sexual dimorphism (although see the comment below on clarifying exactly which results are novel to this paper). But the contribution of the paper to testing major hypotheses for a negative intersex correlation, particularly hypothesis 2, is relatively minor.

- An additional issue is that it is difficult to tell where the authors are presenting novel results and where they are recapitulating results from the previous literature. For example, Section 3.2.1 and 3.2.1.1, which describe how the intersex genetic correlation constrains the rate, but not the outcome, of sexually discordant selection in the infinitesimal limit replicates results originally derived in Lande 1980, as noted by the authors, and has been extended in later papers (e.g. for comparative phylogenetics in Chevrud et al. 1985, for the multivariate case in Wyman et al. 2013). Section 3.0, describing allelic dynamics under a two-sex polygenic model of polygenic adaptation is described in Muralidhar & Coop 2024 (e.g. eq. 8 in this paper corresponds to eq. S7 in that paper). It would improve the readability of this paper if sections re-deriving previously established results were reduced in length, and allow the authors to better highlight the original results they have derived.

Lines 73 -75: These models of sexually antagonistic selection implicitly assume that there is a perfect correlation in a phenotype expressed in both males and females, with the phenotype increasing the fitness of one sex and decreasing the other. This is why sexual conflict - or opposing directional selection in males and females - cannot be resolved in these models. In models of sexually antagonistic conflict where this assumption is violated - e.g. by the evolution of sex-specific expression - this does not hold.

Lines 260-262, 291-292: The width of the selective function, relative to the square root of the additive genetic variance in the trait, is often used empirically to describe the strength of stabilizing selection. I understand the appeal of parameterizing based on the average fluctuation from the optimum, but it would be helpful if the authors could provide the strength of stabilizing selection in these terms, to give greater intuition for the strength of selection in these regimes.

Lines 332-335: A minor point, but Lande 1980 and Reeve & Fairburn 2001 are specifically considering models in which the male optimum increases due to female mate preferences, which is why this directionality in quantifying sexual dimorphism is useful.

Lines 787-802: The authors differentiate between 'convergent' and 'divergent' shifts in phenotypic optima. However, both types of shifts induce opposing directional selection on males and females - that is, 'sexually discordant' selection in the terminology of Cheng & Houle 2020. It would be helpful if the basic commonality between these types of shifts were made more clear here.

Lines 445-457: The effect of a sex-specific mutation on the female or male trait seems to be of magnitude 1, while the effect of shared mutations on the female and male trait is of magnitude (approx.) 0.707. All else being equal, under stabilizing selection, would the shared mutations not experience greater under-dominant selection at equilibrium due to their higher averaged effect sizes? Is there then a reason to make this parameter choice in the simulations? In particular, could this affect the rate of equilibration from the simulation results? (lines 979-989)

Lines 998 - 1010, 1072-1077: This discussion section seems to be conflating definitions of genetic variants based on their phenotypic effects (sex-biased) with their fitness effects (sexually antagonistic). Mutations with effects in each sex of different magnitude, for example, which are here stated to be very rare, are a common form of sex-biased phenotypic effects in humans for example (Zhu et al. 2023).

Lines 949-957: The authors may be interested in Connor 2012, which suggests that genetic correlations are not a useful measure of evolution constraint in quantitative traits, with a similar logic to that presented here.

Code: No simulation code is available at the link provided in the paper.

Figure S2: Are there error bars in this image? The caption states there are.

Cheng, C., & Houle, D. (2020). Predicting multivariate responses of sexual dimorphism to direct and indirect selection. *The American Naturalist*, 196(4), 391-405.

Conner, J. K. (2012). Quantitative genetic approaches to evolutionary constraint: how useful?. *Evolution*, 66(11), 3313-3320.

do O, I., & Whitlock, M. C. (2023). The evolution of genetic covariance and modularity as a result of multigenerational environmental fluctuation. *Evolution letters*, 7(6), 457-466.

Muralidhar, P., & Coop, G. (2024). Polygenic response of sex chromosomes to sexual antagonism. *Evolution*, 78(3), 539-554.

Poissant, J., Wilson, A. J., & Coltman, D. W. (2010). Sex-specific genetic variance and the evolution of sexual dimorphism: a systematic review of cross-sex genetic correlations. *Evolution*, 64(1), 97-107.

Reeve, J. P., & Fairbairn, D. J. (2001). Predicting the evolution of sexual size dimorphism. *Journal of Evolutionary Biology*, 14(2), 244-254.

Walsh, B., & Blows, M. W. (2009). Abundant genetic variation+ strong selection= multivariate genetic constraints: a geometric view of adaptation. *Annual review of ecology, evolution, and systematics*, 40(1), 41-59.

Wyman, M. J., Stinchcombe, J. R., & Rowe, L. (2013). A multivariate view of the evolution of sexual dimorphism. *Journal of Evolutionary Biology*, 26(10), 2070-2080.

Zhu, C., Ming, M. J., Cole, J. M., Edge, M. D., Kirkpatrick, M., & Harpak, A. (2023). Amplification is the primary mode of gene-by-sex interaction in complex human traits. *Cell Genomics*, 3(5).

We thank the reviewers for their thoughtful and constructive feedback. Below, we provide a point-by-point response to each comment. Reviewer/editor comments appear in black, and our responses are provided in red. We refer to changes in the clean (C) and marked (M) versions of the manuscript using the format L<line> p<page> (C/M). For example, L31-56 p3 (C) / L59-78 p3 (M) refers to lines 31 to 56 of page 3 in the clean version and lines 59 to 78 in the marked version.

Dear Dr. Hayward:

Two experts in the field have reviewed your manuscript, and I have read it as well. While your manuscript is not currently acceptable for publication in GENETICS, we would welcome a substantially revised manuscript. Both reviewers have comments and concerns to be addressed in a revised manuscript. You can read their reviews at the end of this email.

Both reviewers have concerns that should be addressed. Reviewer 2 in particular raises important concerns about the extent to which the authors test the hypotheses they lay out as their motivation and the extent to which previous papers have already addressed these concerns. We look forward to receiving your revised manuscript. Please let the editorial office know approximately how long you expect to need for revisions.

Upon resubmission, please include:

1. A clean version of your manuscript;
2. A marked version of your manuscript in which you highlight significant revisions carried out in response to the major points raised by the editor/reviewers (track changes is acceptable if preferred);
3. A detailed response to the editor's/reviewers' feedback and to the concerns listed above. Please reference line numbers in this response to aid the editor and reviewers.

Your paper will likely be sent back out for review.

Additionally, please ensure that your resubmission is formatted for GENETICS <https://academic.oup.com/genetics/pages/general-instructions>

We have done so. Both the clean and marked versions of the manuscript have been formatted according to the GENETICS guidelines. In this regard, among other modifications, we have eliminated the numbering of the sections and adapted the cross-references between sections accordingly (using the \nameref command). We have also revised the figures so that panel labels now use lowercase letters.

We would like to note one minor issue with the submission system. While both the GENETICS website and the submission instructions specify a 100-word limit for the Article Summary, the submission portal only allows 80 words in the entry field. Accordingly, we included a shortened version (80 words) in the system to comply with the field restriction, while retaining the full 100-word version in the manuscript files.

Follow this link to submit the revised manuscript: Link Not Available

Sincerely,

Graham Coop
Associate Editor
GENETICS

Approved by:
Sarah Otto
Senior Editor
GENETICS

Reviewer #1 :

This manuscript explores the theoretical relationship between sexual dimorphism (SD) and intersex genetic correlation under models of sex-specific stabilizing selection, mutation, and genetic drift. The authors critically examine the intuitive expectation of a negative correlation between SD and intersex genetic correlation by analyzing equilibrium and transient evolutionary dynamics across different genetic architectures (infinitesimal vs. multigenic).

Overall, the results are well-structured, comprehensive, and thoroughly discussed. Two particularly interesting findings are:

1. Convergent sexually dimorphic adaptation may lead to a positive association between intersex correlation and SD.
2. Genetic drift alone can generate non-zero absolute SD.

These are indeed the two of the main contributions of our study, and we thank the reviewer for expressing them so explicitly. We have modified the manuscript to more effectively emphasize these two findings.

With these results, I have two key thoughts:

1.) I'm just guessing - a strong intersex genetic correlation may buffer the effect of genetic drift, potentially reducing SD when sex-specific optima are similar. This could contribute to the commonly observed negative relationship between intersex correlation and SD. I would be interested in further analysis of this effect-specifically, how intersex correlation influences non-zero SD generated by drift-as it may provide a mechanistic explanation for the negative association. However, I leave this to the editors' and authors' discretion.

This was also our intuition. When the two optima are close, we expected SD to be lower when the intersex correlation was very high. And we initially actually found this to be the case in our simulations! Unfortunately, the effect turned out to be a result of the choice of initial condition (both sexes having equal mean phenotypes—of zero—so initial $SD = 0$) combined with running insufficiently long burn-ins. We were running our simulations for a burn in of $10N$ generations,

when (for some parameter choices) a burn in of 100 or even 500N generations is required for the SD to equilibrate if intersex correlation is very high. As it turns out, when intersex-correlation is high, the autocorrelation of SD decays extremely slowly with the two mean phenotypes drifting in a coordinated way (we have tried to illustrate this in Fig S6). However, this slow rate of change does not turn out to affect the steady-state level of SD. We found that the steady-state mean SD is in fact the same for all levels of intersex correlation < 1 (shown, for example, in Figs 1 and S3) and is always equal to $2\sqrt{2/\pi} \cdot \delta \approx 1.6 \cdot \delta$ when the difference in optima is small (Figs 1b and S3b).

2.) Under fluctuating selection around a mean optimum, non-zero SD may emerge when the mean optima for both sexes coincide. In such a scenario, I hypothesize that intersex correlation would negatively correlate with SD. This could be the basis for a separate study. Personally, I find the interplay between genetic drift, intersex correlation, and fluctuating selection particularly intriguing and worth exploring in future work to better explain the relationship between rfm and SD.

This is an interesting line of inquiry. Our intuition is actually that intersex correlation probably would not negatively correlate with SD in that scenario. Since selection is concordant and SD is not being selected to either increase or decrease, we would expect it to behave fairly similarly to steady-state (drifting very slowly). However, this is just waving hands around, and we would also be most interested to see future work in this direction!

Specific Comments:

Lines 744/ 813: The assumption that multigenic architectures are less constrained than the infinitesimal model is supported by literature on G-matrix evolution. The genetic covariance structure is generally more stable under the infinitesimal model (e.g., Barton & Turelli, 1987, 1989; Lande, 1979) than under a more Mendelian-like architecture, resulting in stronger constraints under the infinitesimal model. For further discussion and related empirical evidence, see the discussion section of Cai et al. Dissecting genetic correlation and pleiotropy through a genetic cross.

We thank the reviewer for raising this point, which provides more context to our results. We included it in the Discussion (L85-89 p20 (C) / L9-14 p23 (M)).

Genetic Architecture: Why were these two genetic architectures (infinitesimal vs. multigenic) chosen for comparison? How does the multigenic model relate to the classical "House of Cards" (HoC) model? Are they conceptually similar? It would be beneficial to mention the HoC model in the introduction to clarify its relationship with the frameworks used in this study.

This is a good point; readers familiar with the classical quantitative genetics literature are likely to wonder about this. We have renamed the section "Parameterizing the allelic effects" to "Specifying the genetic architecture" (starting at L21 p7 (C) / L93 p7 (M)), moved it, and revised it so that it now functions to properly explain our choice of genetic architectures (paragraph on L33-49 p7 (C)/L7-23 p8 (M)). In addition, we now compare the multigenic genetic architecture

to classical House of Cards models (paragraph on lines L50-67 p7 (C)/L24-41 p8 (M)), which we include here for convenience:

It is helpful to compare this approach to classical work in quantitative genetics, where less infinitesimal trait architectures are typically captured using a House of Cards model, which assumes that mutations replace the existing allelic effect at each locus with a new, randomly drawn value (Turelli 1984; Burger et al. 1989; Zhang and Hill 2003). These models typically make three key assumptions. First, that a continuum of alleles is possible at each of a fixed number of loci (in contrast, we use a bi-allelic, infinite sites model). Second, that at each locus selection dominates mutation (by using an infinite sites model, we also implicitly make this second assumption). Third, that all alleles are subject to strong selection ($2Ns_e \gg 1$). It is important that we are able to relax this third assumption, both because distributions of new mutations completely lacking in nearly neutral or weakly selected alleles seem unlikely, and because weakly selected alleles play an important role in long-term dynamics following a shift in the optimum, even with a multigenic genetic architecture (Hayward and Sella 2022).

Line 947 repeated words: 'similar'

Thank you for spotting this. The sentence with the original mistake (in L99 p20 (M)) has been replaced with a new one in the revised version.

There is no code in the provided github site.

The code has now been uploaded to the github site.

Cheers,
Haoran Cai

Reviewer #2 :

This paper examines whether the evolution of sexual dimorphism in a quantitative trait is associated with a negative intersex genetic correlation in that trait. Specifically, this paper analyzes models of a polygenic trait, along with simulations of those models, to test different hypotheses for why there should be a negative association between the intersex genetic correlation and the level of phenotypic divergence between males and females. While there are interesting results in this paper, I have major concerns about whether the authors are actually able to test their hypotheses for the association between sexual dimorphism and the intersex genetic correlation. These, and other comments, are detailed below.

--- The authors state that they will test two hypotheses that can explain negative correlation between the level of sexual dimorphism and the intersex genetic correlation. The first posits that a high intersex correlation means that the rate of adaptation in a sexually dimorphic direction (relative to a sexually concordant direction) is slower, although, given an imperfect genetic correlation, the different male and female optima will eventually be reached. This is, as the authors note, originally stated in Lande 1980, and very well established in the quantitative genetic literature (e.g. Poissant et al. 2010, Wyman et al. 2013) in the context of

sexual dimorphism, and more broadly, in the context of constraints on multivariate adaptation (Walsh & Blows 2009). The authors confirm that the rate of adaptation in a sexually dimorphic direction is constrained by the intersex correlation, and therefore confirm hypothesis 1.

This comment refers to what we discuss as the “first hypothesis”—one of two mechanisms frequently cited in the literature as potentially generating a negative association between intersex genetic correlation and sexual dimorphism. As we elaborate in response to this reviewer’s third comment, we acknowledge that we do not directly test either hypothesis. Rather, we use models of sex-specific stabilizing selection to explore the general conditions under which a correlation between intersex genetic correlation and sexual dimorphism might emerge. To our knowledge, this has not been explicitly examined in theoretical work, despite clear verbal expectations of a negative association expressed in the literature on sex-specific adaptation. It is precisely because such expectations are widespread, and because two mechanisms are commonly invoked to explain them, that we find it useful to contextualize our study in terms of these prior ideas.

With this in mind, and more directly addressing the reviewer’s concern regarding our treatment of “hypothesis 1”, we do recapitulate the classical result that intersex genetic correlation constrains the rate of evolution of sexual dimorphism, because it is crucial to understanding the relevance of our findings. We clearly acknowledge that this result is well established, citing the relevant literature throughout the manuscript. Importantly, we also emphasize how this classical result has shaped prevailing intuition in the field—specifically, the expectation that such a constraint should produce a negative association between intersex correlation and sexual dimorphism (i.e., hypothesis 1).

In light of this, one of the key contributions of our study is to show that, when this classical result is placed in a broader evolutionary context, it does not actually confirm hypothesis 1. In particular, we demonstrate that the fact that intersex genetic correlation constrains the rate of adaptation could just as plausibly support a **positive** association between sexual dimorphism and intersex correlation. For the constraint to result in a **negative** association, an **additional assumption** must be met: that changes in sex-specific optima more frequently favor increased rather than decreased dimorphism. As far as we are aware, there is no empirical or theoretical basis to assume this.

We apply a similar logic to “hypothesis 2” (see our response to the next comment), and therefore conclude that both commonly cited mechanisms could, under realistic conditions, generate either a negative or **positive** association. To our knowledge, this has not previously been acknowledged.

We recognize that this point was not emphasized in the original manuscript, and we have revised the text extensively to clarify it. Specifically:

1. We have rewritten the Article summary and significantly revised the Abstract to emphasize that the two mechanisms most often proposed to explain a negative association can also give rise to a **positive** one.
2. We are now more explicit about this insight throughout the manuscript. For example, in the Introduction (starting at L80 p3 (C) / L124 p3 (M)), we state:

By considering the transient phase of adaptation to new sex-specific optima (during which directional selection acts), we illustrate that mechanisms underlying the two extensively-discussed hypotheses to explain a negative association between intersex correlation can both generate a relationship between the two, even with a non-evolving genetic architecture. Crucially, however, we show that the association generated is only negative if adaptation more frequently favours increased dimorphism over decreased dimorphism, i.e., if divergent shifts in optima, which increase the distance between sex-specific optima, are more common than convergent shifts which decrease the distance (see Box 1 for a more detailed explanation of the terminology). Indeed, we find that if convergent shifts are more common than divergent shifts the same two mechanisms can generate a positive association between sexual dimorphism and intersex correlation. This is important because it exposes a hidden assumption behind the prevailing intuition: namely, that divergent shifts are consistently favoured over convergent shifts. To our knowledge, there is no reason to expect that this should be the case.

3. In the Results section, we have made numerous modifications both in response to this and later comments, and also clarifying which results are original and where we are recapitulating established findings. Details of these revisions can be found in our response to this reviewer's third comment.
4. In the Discussion, we explicitly address the assumption that divergent shifts are more common than convergent ones and explain why this assumption does not necessarily hold (L66–74 p20 (C) / L109–119 p22 (M)).

The second hypothesis is that the genetic basis of the intersex genetic correlation will itself evolve in response to repeated bouts of discordant selection in males and females. In quantitative genetic terms, the genetic variance-covariance matrix, or the underlying mutational structure giving rise to that matrix, can evolve such that the intersex genetic correlation is reduced (a similar logic has been applied to explain how fluctuating selection can drive the evolution of a modular genetic architecture, see, e.g., do O & Whitlock 2023). But the authors do not test this hypothesis, because they assume that the genetic architecture of the trait stays constant. As would be expected under this regime, and as noted in prior papers (Reeve and Fairbairn 2001, Wyman et al. 2013), there are only transient changes in the intersex correlation under these assumptions, because the intersex genetic correlation is essentially prevented from evolving.

This comment refers to what we discuss as the second hypothesis—often cited in the literature as an alternative to the first: that a negative association between intersex genetic correlation and sexual dimorphism could arise because the intersex correlation itself evolves in response to sex-specific selection. We recognize that an evolving genetic architecture is likely the scenario that Lande (and others) had in mind when suggesting that intersex correlation should decrease during the evolution of sexual dimorphism, and we provide this context (e.g., L71–90 p2 (C) / L20–39 p3 (M) in the Introduction).

While acknowledging this, we still believe there is significant value in examining this relationship under the assumption of a non-evolving genetic architecture. We now provide a more detailed justification for this modeling choice in the Introduction (L46–66 p3 (C) / L87–107 p3 (M)):

We consider these genetic architectures to be non-evolving (i.e. we are not considering modifier loci that could lead to stable decreases in intersex covariances). While this likely excludes certain mechanisms that might contribute to stable reductions in rfm during sexual dimorphism evolution, as suggested by the second hypothesis above, we make this choice for four reasons. First, and most importantly, it is the natural first step: we cannot hope to understand the relationship between intersex correlation and sexual dimorphism in the most general setting without first understanding their co-evolutionary dynamics with a non-evolving genetic architecture. This is particularly important given that some of our findings with a non-evolving architecture are unexpected. Second, the evolution of intersex covariances is expected to be a slow process, so our assumptions are likely to reflect the dynamics of shorter-term evolutionary processes (Williams and Carroll 2009; Stewart et al. 2010). Third, our results are more directly comparable to those of most prior studies, which have also assumed a non-evolving genetic architecture (Lande 1980; Reeve and Fairbairn 2001; Wyman et al. 2013). Fourth, some of our conclusions are expected to be robust to relaxing this assumption (see the Discussion for more details).

We have also substantially revised the Discussion to provide clearer context for how our results relate to classical models, and to justify our choice of assuming a fixed genetic architecture (L69 p19 to L14 p20 (C) / L41 p21 to L19 p22 (M)). In particular, we now explicitly outline how our modeling framework could be extended to incorporate an evolving genetic architecture (L15–33 p20 (C) / L29–47 p22 (M)), which we agree would be a valuable direction for future work.

Importantly, we also discuss why a central insight of our study—that the two prevailing hypotheses can lead to either negative or **positive** associations—may well hold even under an evolving genetic architecture. While this extension would undoubtedly add complexity, we expect the core logic and interpretive framework to remain relevant.

Given the previous work establishing hypothesis 1 in very similar regimes to those the authors consider, and the fact that the authors can only test a very limited version of hypothesis 2 due to their modelling assumptions, it is difficult to see why the paper is framed in terms of testing these two hypotheses at all.

I think the results of the authors detailing the evolutionary dynamics of sexual dimorphism and the intersex genetic correlation under a multigenic, rather than infinitesimal, genetic architecture, could present an interesting twist on the existing literature on the evolution of sexual dimorphism (although see the comment below on clarifying exactly which results are novel to this paper). But the contribution of the paper to testing major hypotheses for a negative intersex correlation, particularly hypothesis 2, is relatively minor.

We thank the reviewer for prompting us to reconsider how we frame the manuscript and communicate its contributions. This feedback motivated us to revise the way we introduce the

paper, explain our analytical approach, and contextualize our findings, to better convey the relevance and novelty of our work.

Our motivation for this study stemmed from the surprising prevalence of the expectation that there should be a negative correlation between intersex genetic correlation and sexual dimorphism—despite the absence of any formal theoretical investigation of whether the verbal hypotheses commonly invoked to support this expectation actually hold. This is what we set out to do, and to this end it is vital to revisit the classical results that have been results that have shaped prevailing intuition in the field.

Crucially, and as we failed to clearly convey in the original submission, our findings do not confirm either of the two hypotheses. This is, in our view, a central contribution: we use (and extend) established models to **challenge established intuition**. As we detail in our response to this reviewer's first comment, we have revised the manuscript to make this point much more explicit. We also now emphasize how our findings could explain why the expected negative association is only inconsistently observed in nature and illustrate the importance of critically re-evaluating long-standing assumptions.

In line with this, we have revised the manuscript to avoid referring to our study as “testing” these hypotheses. Instead, we explain that we contextualize our results in terms of how prior literature has typically framed the relationship between intersex correlation and sexual dimorphism—namely, through expectations derived from these two hypotheses. Relevant changes can be found in L18–31 p11 (C) / L21–38 p12 (M), L45–54 p17 (C) / L75–87 p18 (M), and L111–118 p18 (C) / L54–62 p20 (M).

As one representative example, we have substantially rewritten the concluding paragraph of the Introduction to clarify our framing. In L1–14 p4 (C) / L69–82 p4 (M), we now state:

Altogether, in this study we take classical results and well-established expectations about the coevolutionary dynamics between sexual dimorphism and intersex correlation as the starting point. We re-examine these results from a new perspective, formally articulating the commonly accepted reasoning behind the expectation of a negative correlation between the two. Our analysis challenges prevailing intuition by uncovering the implicit assumptions underlying these arguments, thereby highlighting the importance of clearly stating the assumptions and mechanisms that underpin widely held hypotheses. Moreover, we show how established results integrate into a broader mathematical framework, providing a more complete description of the evolutionary dynamics of a trait under sex-specific stabilizing selection, both at and away from equilibrium.

As also noted in our response to the first comment, we now stress that our results **do not necessarily support the expectations** derived from prior intuition, which we believe is a key conceptual contribution of the paper.

In addition, we highlight a second conceptual insight that emerged from our analyses—also acknowledged by Reviewer 1: **that genetic drift can generate non-zero sexual dimorphism even when male and female optima are aligned**. We have added or expanded on this point in several places, including the Abstract, in L67–79 p3 (C) / L108–123 p3 (M) of the

Introduction, and in the Discussion section titled “Drift generates a nonzero SD even when sex-specific optima are aligned” (starting at L56 p18 (C) / L109 p19 (M)).

- An additional issue is that it is difficult to tell where the authors are presenting novel results and where they are recapitulating results from the previous literature. For example, Section 3.2.1 and 3.2.1.1, which describe how the intersex genetic correlation constrains the rate, but not the outcome, of sexually discordant selection in the infinitesimal limit replicates results originally derived in Lande 1980, as noted by the authors, and has been extended in later papers (e.g. for comparative phylogenetics in Chevraud et al. 1985, for the multivariate case in Wyman et al. 2013). Section 3.0, describing allelic dynamics under a two-sex polygenic model of polygenic adaptation is described in Muralidhar & Coop 2024 (e.g. eq. 8 in this paper corresponds to eq. S7 in that paper). It would improve the readability of this paper if sections re-deriving previously established results were reduced in length, and allow the authors to better highlight the original results they have derived.

We thank the reviewer for raising this point. We agree that it is critical to clearly distinguish between previously established results and those that are novel to this study. To address this, we firstly made many modifications to clearly convey that certain aspects of our findings are novel because they challenge rather than confirm established intuition (described in response to previous comments). We also made numerous revisions to improve transparency throughout the paper.

In particular, in the Introduction, we now provide a concise summary of our novel contributions (L99–124 p3 (C) / L43–68 p4 (M)). For convenience, we reproduce the paragraph here:

Additionally, in the course of our investigation into the coevolution of sexual dimorphism and intersex correlation, we examine in detail the dynamics of sex-specific adaptation under stabilizing selection, mutation and drift, with a highly polygenic genetic architecture. Incorporating the effects of genetic drift, we derive novel expressions for sex-specific variances, the covariance between sexes, intersex correlation, and sexual dimorphism at equilibrium. We further analyze how the phenotypic response to a shift in the optimum arises from allele frequency dynamics, extending the framework of Muralidhar & Coop (2024)---which is limited to genetic architectures where predictions from the infinitesimal limit hold---and generalizing the single-sex results of Hayward & Sella (2022). Regarding the response of sex-specific means, we delineate the conditions under which deviations from Lande’s classical predictions become appreciable. While previous studies (e.g., Reeve & Fairbairn, 2001) have discussed such deviations in terms of changes to (co)variances, we demonstrate that third-order central moments of the phenotypic distribution---which emerge in our generalization of the two-sex breeder’s equation---also play a critical role, particularly after the initial rapid phase. Finally, we characterize the long-term equilibration process by providing approximations for the rate at which the component of the mean phenotype maintained by fixations, rather than segregating variation, converges to the new optimum---a description, to our knowledge, not previously offered in two-sex models.

We have also revised individual Results sections to more clearly indicate where we are recapitulating previous work, where we extend it, and where we present entirely new results. Specific actions taken are listed below. In some cases, we have chosen to retain detailed derivations or discussions—for clarity and accessibility—and also explain those decisions below.

1. **The initial part of the results mentioned by the reviewer (L10 p6 (C) / L84 p6 (M), formerly section 3.0):**

The first part of this section sets up the precise definitions of intersex correlation, as well as that of signed and absolute sexual dimorphism (L10-61 p6 (C) / L84 p6 to L25 p7 (M)), which are crucial for understanding the rest of the paper. This is followed by a brief overview of what is contained in the Results (L62-81 p6 (C) / L26-55 p7 (M)). Equation 8, which comes next, is now explicitly referenced to Muralidhar & Coop (2024), as suggested. Although this equation appears in their SI, our use of it supports novel results, including our generalization of the two-sex breeder's equation (Equation 35). We retained explanatory material below Equation 8 (L96 p6 to L20 p7 (C) / L70-91 p7 (M)) for clarity and self-containment but added citations at L8 & L13 p7 (C) / L79 & L84 p7 (M).

2. **“The relationship between rfm and SD at equilibrium” (L91 p7 (C) / L65 p8 (M), formerly Section 3.1):**

The first part of this section establishes the independence of intersex correlation and sexual dimorphism in the absence of drift induced fluctuations in the mean phenotype. In the initial submission, most of this section was taken up by “The intersex correlation at equilibrium” (formerly Section 3.1.1.2), and we have revised the paper to make it clear that the **conclusions** that we make in this section have been previously established. However, the **analytical expressions** that we present (derived using a diffusion approximation) for the equilibrium sex-specific variances, covariance and intersex correlation are new. We now state this and also give readers who are less interested in mathematical results a clear path to skip this subsection (L21-27 p8 (C) / L3-10 p9 (M)). This required introducing a little notation in the simulations section that readers skipping the maths might have missed (L87-91 p5 (C) / L39-43 p6 (M)).

3. **Sections on drift-induced dynamics (formerly 3.1.2, 3.1.3, and 3.1.2.1):**

All of our results accounting for (and discussing) drift-induced fluctuations in the mean are novel. We have therefore left these parts of the manuscript largely unchanged: “Drift generates nonzero E[SD] even when sex-specific optima coincide” starting line L20 p9 (C) / L12 p10 (M), and formerly Section 3.1.2; “Drift does not induce an association between E[SD] and rfm” starting line L50 p10 (C) / L48 p11 (M) and formerly Section 3.1.3; and “The significance of drift-inflated SD” starting line L70 p10 (C) / L48 p11 (M), formerly Section 3.1.2.1.

4. **“A negative relationship between rfm and SD -- exploring common hypotheses” (L14 p11 (C) / L17 p12 (M), formerly Section 3.2):**

In points 5 and 6 below we discuss the first two subsections of this section, which are specifically mentioned by the reviewer.

5. **“Exploring H1: rfm determines the rate of SD evolution” (L54 p11 (C) / L61 p12 (M), formerly section 3.2.1):**

We now make it clear that while we confirm Lande’s result—that intersex correlation constrains the rate of sexual dimorphism evolution—our interpretation of this result is different from that of previous authors: we show that Lande’s result could just as plausibly be used to argue against hypothesis 1 (L95-99 p11 (C) / L15-21 p13 (M)).

6. **“Adaptation in the infinitesimal limit: rfm determines the relative rate of sexually-concordant vs sexually-discordant evolution”: (L100 p11 (C) / L22 p13 (M), formerly Section 3.2.1.1):**

We start by showing that the two-sex extension of the breeders equation is a special case of our more general expression which incorporates third moments and, so far as we are aware, is new (Equation 35). We then introduce the change of variables—Lande’s change of variables as we make clear—which we use not only in this section but for the remainder of the paper and we therefore believe is necessary (unlike Lande we use the change of variables not just for the distance from the optimum, but also for third moments and the fixed background). Between lines L37 p12 and L20 p13 (C) / L65 p13 and L24 p15 (M) we rederive Lande’s results with full attribution, and point out that, as our simulations confirm, they fit well with our approximately infinitesimal genetic architecture. We need the reader to thoroughly understand these results for two reasons 1) so that our later discussion of deviations from his predictions when there is a nonzero third moment in with a multigenic genetic architecture makes sense, and 2) so that our explanation regarding why the results do not necessarily lead to the negative correlation they are typically taken to imply, makes sense. We believe that simply presenting the expressions for the length of the rapid phase for sexually concordant and discordant adaptation (Equations 43 and 44) and for signed sexual dimorphism (Equation 45), would considerably diminish the accessibility of the paper for readers unfamiliar with the classical results.

7. **“Adaptation with a multigenic genetic architecture: transient changes in the 2nd and 3rd order moments of the phenotype distribution alter the dynamics of phenotypic adaptation” (L15 p14 (C) / L40 p15 (M), formerly section 3.2.1.2):**

This section considers how short-term dynamics are affected by a multigenic architecture. Other studies (eg. Reeve & Fairbairn, 2001) have looked at deviations of the infinitesimal regime by considering changes in the (co)variances, which we acknowledge (L114 p3 (C) / L58 p4 (M)). We extend their analysis both conceptually and mathematically.

8. **Higher intersex correlation delays equilibration for sex differences (L13 p16 (C) / L41 p16 (M), formally 3.2.1.3):**

So far as we know, in the two-sex case, the rate of equilibration has not previously been quantified using the decay of distance of the fixed background from the optimum. Therefore our equilibration results which focus on determining this rate of the decay in the distance of the fixed background from the optimum are novel for both genetic architectures, which (in the revised manuscript) we point out in the introduction (L119-124 p3 (C) / L63-68 p4 (M)). Indeed, with an approximately infinitesimal genetic architecture phenotypic variance does not increase during the rapid phase and the third-moment is always close to zero, so only a measure that accounts for fixation

(such as the distance of the fixed background from the optimum) can be used to determine if equilibrium has been re-established. We are not aware of other such measures existing in the literature.

9. **“rfm and SD might negatively or positively correlate” (L6 p17 (C) / L29 p18 (M), formerly section 3.2.1.4):**

Formerly titled “H1 holds -- given an additional assumption”, we renamed this section and revised the text to clearly convey that we have actually discovered that the argument used to support H1 could support either a negative or positive correlation between rfm and SD.

10. **“Exploring H2: sex-specific directional selection transiently reduces rfm” (L44 p17 (C) / L74 p18 (M), formerly section 3.2.2):**

In this section we describe how the transient decrease in rfm during sexually-discordant evolution might contribute to an association between rfm and SD, in a similar logic to that suggested by H2 – although, as we illustrate, this association might be positive as well as negative. While the transient decrease in rfm in SD evolution with a multigenic genetic architecture has been described before (Reeve & Fairbairn 2001), they did not discuss its potential to generate an association between rfm and SD, or provide conditions for when this decrease becomes appreciable (i.e., with our multigenic architecture), which we do here. In the revised manuscript, we fully discuss their contribution and our extensions to their work in the discussion in L69-88 p19 (C) / L50-71 p21 (M).

Lines 73 -75: These models of sexually antagonistic selection implicitly assume that there is a perfect correlation in a phenotype expressed in both males and females, with the phenotype increasing the fitness of one sex and decreasing the other. This is why sexual conflict - or opposing directional selection in males and females - cannot be resolved in these models. In models of sexually antagonistic conflict where this assumption is violated - e.g. by the evolution of sex-specific expression - this does not hold.

It is true that if the single locus can evolve to have sex-specific expression, then sexual conflict can be resolved even in the single-locus model. We replaced “sexual antagonism” with “sex-specific selection” to avoid confusion and we have added a clarification to this effect: “unless the locus can evolve to have sex-specific effects” (L50-51 p2 (C) / L90 p2 (M)).

Lines 260-262, 291-292: The width of the selective function, relative to the square root of the additive genetic variance in the trait, is often used empirically to describe the strength of stabilizing selection. I understand the appeal of parameterizing based on the average fluctuation from the optimum, but it would be helpful if the authors could provide the strength of stabilizing selection in these terms, to give greater intuition for the strength of selection in these regimes.

Good idea. We now do this in L93-98 p5 (C) / L45-50 p6 (M).

Lines 332-335: A minor point, but Lande 1980 and Reeve & Fairburn 2001 are specifically considering models in which the male optimum increases due to female mate preferences, which is why this directionality in quantifying sexual dimorphism is useful.

We now explicitly state this in L40-44 p6 (C) / L4-8 p7 (M).

Lines 787-802: The authors differentiate between 'convergent' and 'divergent' shifts in phenotypic optima. However, both types of shifts induce opposing directional selection on males and females - that is, 'sexually discordant' selection in the terminology of Cheng & Houle 2020. It would be helpful if the basic commonality between these types of shifts were made more clear here.

In our initial submission, we referred to selection where the difference between sex-specific optima changes as 'sexually-dimorphic' selection. This was explained in box 1, but it should have been addressed in the text itself too. We have now added a brief in-text explanation the first time the term is mentioned in L34-36 p2 (C) / L73-75 p2 (M) along with a reference to the box. We also think that 'sexually-discordant' is clearer terminology than 'sexually-dimorphic' (as the opposite of sexually-concordant adaptation) and we have changed this throughout the manuscript.

At the lines that were formally 787-802, which the reviewer is referring to in the present comment, we have added a few extra words to help remind the reader of the commonality between the two types of shifts at this point in the manuscript, as well as another link to the box (L26 p17 (C) / L55 p18 (M)).

Lines 445-457: The effect of a sex-specific mutation on the female or male trait seems to be of magnitude 1, while the effect of shared mutations on the female and male trait is of magnitude (approx.) 0.707. All else being equal, under stabilizing selection, would the shared mutations not experience greater under-dominant selection at equilibrium due to their higher averaged effect sizes? Is there then a reason to make this parameter choice in the simulations? In particular, could this affect the rate of equilibration from the simulation results? (lines 979-989)

This is an interesting issue and we initially spent quite some time deliberating over this choice. We do not think it is obvious what one should hold constant when comparing the effect of differing levels of intersex correlation between different traits (which might after all be measured in different units).

We found it especially important that the alleles that underlie the traits being compared segregate at the same frequencies. And we ultimately decided it was important to hold the overall strength of selection on alleles constant (as well as the genetic variance relative to V_S). This choice was motivated largely by the fact the exact choice of distribution of the overall strength of selection on alleles (the distribution of $2Na^2/V_S$) has extremely clear interpretable effects on the dynamics that precisely mirror those in the single-sex case, and changing this distribution as we change r_{fm} would remove our ability to observe this.

One example of this is in the lower panel of Fig 2f showing the multigenic case (referred to in lines 979-989 mentioned by the reviewer), the precise shape of the trajectory of the distance of the fixed background from the optimum as it goes first slower than $\text{Exp}(-t/(2N))$ and then faster is precisely determined by the choice of distribution of scaled selection strengths. This is why all the lines approximately overlap.

With a different choice of parameterization the precise way in which the distribution of scaled selection coefficients tunes the dynamics would not be evident. However, we do not think that the choice has a qualitative effect on the simulation results referred to. Specifically, it does not alter 1) the observation that equilibration occurs on the order of $2N$ generations. 2) the observation that equilibration can be slower for a high r_{fm} but the degree of slow down is small relative to the slow down over the short-term.

Lines 998 - 1010, 1072-1077: This discussion section seems to be conflating definitions of genetic variants based on their phenotypic effects (sex-biased) with their fitness effects (sexually antagonistic). Mutations with effects in each sex of different magnitude, for example, which are here stated to be very rare, are a common form of sex-biased phenotypic effects in humans for example (Zhu et al. 2023).

We thank the reviewer for pointing this out. We have addressed the concern in two ways. . First, we have simply removed the words “sex-biased” and “sexually-antagonistic” to avoid confusion (L101-109 p19 (C) / L91-101 p21 (M)). Second, we now acknowledge that mutations with sex-differentiated phenotypic effects are common. We implemented both changes in L122 p20 to L10 p21 (C) / L55-73 p23 (M).

Lines 949-957: The authors may be interested in Connor 2012, which suggests that genetic correlations are not a useful measure of evolution constraint in quantitative traits, with a similar logic to that presented here.

We appreciate the reference, as its logic is indeed similar. We expanded the discussion section in which we comment on the limiting view of genetic correlations as evolutionary constraints with a couple more references in L22-31 p21 (C) / L87-198 p23 (M).

Code: No simulation code is available at the link provided in the paper.

The code has now been uploaded to the github site.

Figure S2: Are there error bars in this image? The caption states there are.

The error bars are present but smaller than the width of the markers. We have added a comment in the caption to this effect.

Cheng, C., & Houle, D. (2020). Predicting multivariate responses of sexual dimorphism to direct and indirect selection. *The American Naturalist*, 196(4), 391-405.

Conner, J. K. (2012). Quantitative genetic approaches to evolutionary constraint: how useful?. *Evolution*, 66(11), 3313-3320.

do O, I., & Whitlock, M. C. (2023). The evolution of genetic covariance and modularity as a result of multigenerational environmental fluctuation. *Evolution letters*, 7(6), 457-466.

Muralidhar, P., & Coop, G. (2024). Polygenic response of sex chromosomes to sexual antagonism. *Evolution*, 78(3), 539-554.

Poissant, J., Wilson, A. J., & Coltman, D. W. (2010). Sex-specific genetic variance and the evolution of sexual dimorphism: a systematic review of cross-sex genetic correlations. *Evolution*, 64(1), 97-107.

Reeve, J. P., & Fairbairn, D. J. (2001). Predicting the evolution of sexual size dimorphism. *Journal of Evolutionary Biology*, 14(2), 244-254.

Walsh, B., & Blows, M. W. (2009). Abundant genetic variation+ strong selection= multivariate genetic constraints: a geometric view of adaptation. *Annual review of ecology, evolution, and systematics*, 40(1), 41-59.

Wyman, M. J., Stinchcombe, J. R., & Rowe, L. (2013). A multivariate view of the evolution of sexual dimorphism. *Journal of Evolutionary Biology*, 26(10), 2070-2080.

Zhu, C., Ming, M. J., Cole, J. M., Edge, M. D., Kirkpatrick, M., & Harpak, A. (2023). Amplification is the primary mode of gene-by-sex interaction in complex human traits. *Cell Genomics*, 3(5).

July 21, 2025

RE: GENETICS-2025-308261

Dear Dr. Hayward:

I am pleased to accept your manuscript titled "The relationship between sexual dimorphism and intersex correlation: do models support intuition?" for publication in GENETICS, pending minor revision. Thank you for your work in response to the reviews.

Please submit your revision along with a brief description of how you modified the manuscript in response to the reviewers' concerns and suggestions (which can be viewed at the bottom of this email). I expect you should be able to submit a revised manuscript within 30 days. A suitably revised manuscript will be acceptable for publication; I don't expect to send it out for review.

Please ensure that you have included a Data Availability Statement at the end of the Materials and Methods section. Details available at <https://academic.oup.com/genetics/content/prep-manuscript>. The DAS should include the accession numbers or DOIs of any data you have placed in public repositories, describe supplemental material, include applicable IRB numbers, and may include specifications for how to properly acknowledge or cite the data.

When revising the ms., please make an effort to shorten it, because that almost always improves a manuscript. We urge authors to heed the advice of Strunk and White: "omit needless words"¹. Follow this link to submit the revised manuscript: Link Not Available

Thank you for submitting this story to Genetics.

Sincerely,

Graham Coop
Associate Editor
GENETICS

Approved by:
Sarah Otto
Senior Editor
GENETICS

Reviewer comments:

Reviewer #1 :

It would be great if authors can make it very clear in the introduction and abstract that you are making assumption of non-evolving genetic architecture during simulation. Hence, the term 'coevolutionary dynamics' of intersex correlation and sexual dimorphism looks a bit weird to me because the intersex correlation is basically constrained by the shared genetic architecture that is set prior to simulation.

When discussing implications, consider the possibility of neutral evolution for sex-specific traits. Under neutrality, a negative correlation between intersex correlation and divergence in sex-specific traits might be expected. Incorporating this scenario, perhaps with a simple simulation, would be beneficial for establishing a null expectation and is relevant to the present results and discussion .

Minor:

Page 1 Line 10 adaption -> adaptation

Fig 3 Caption: dimporhism -> dimorphism

Long sentence Page 18 Line 10 - line 21. confusing.

Reviewer #2 :

This is my second review of the manuscript, and the authors have addressed my previous comments. In particular, the revised framing clarifies the limitations of characterizing the long-term relationship between the intersex correlation and sexual dimorphism under the assumption of a static genetic architecture. I have only minor comments remaining.

P1. Line 85, P2. 58-61. The evolution of genetic architecture is generally slower than the time scale of experimental evolution, but it's unclear why this would be especially relevant for male-female trait architectures compared to any two traits with shared genetic bases (e.g., highly allometric morphological traits). The cited articles do not clarify this distinction, and I'm not aware of any general intuition for it. While this is a minor point, it justifies the authors' decision to focus on non-evolving architectures for testing hypothesis 2, and greater clarity would be helpful.

P19, Lines 110 - 122. The authors may be interested in Carreira et al. 2009, which suggests that the genetic architecture of body size in *Drosophila melanogaster* can evolve greater sex-specificity via mutations at key developmental genes.

Carreira, V. P., Mensch, J., & Fanara, J. J. (2009). Body size in *Drosophila*: genetic architecture, allometries and sexual dimorphism. *Heredity*, 102(3), 246-256.

Figure S4. The horizontal dashed line in part a is not specified in the caption.

Associate Editor comments:

We thank the editor and reviewers for their constructive feedback on our revised manuscript. We have addressed the remaining comments as outlined below, and all changes are included in the revised version of the manuscript (clean and marked). We refer to changes in the clean (C) and marked (M) versions of the manuscript using the format L<line> p<page> (C/M).

Reviewer #1 :

It would be great if authors can make it very clear in the introduction and abstract that you are making assumption of non-evolving genetic architecture during simulation. Hence, the term 'coevolutionary dynamics' of intersex correlation and sexual dimorphism looks a bit weird to me because the intersex correlation is basically constrained by the shared genetic architecture that is set prior to simulation.

We appreciate the reviewer's comment. As also noted by Reviewer 2, we have taken care in the revised manuscript to clearly state and justify our assumption of a non-evolving genetic architecture. This assumption is now discussed early in the Introduction (L49–69 p3 (C) / L50–70 p3 (M)), where we explicitly note: *We consider these genetic architectures to be non-evolving (i.e. we are not considering modifier loci that could lead to stable decreases in intersex covariances)* and discuss the reasons for this choice. We also return to this point in the Discussion, where we further justify this assumption (L104 p19 to L31 p20 (C and M) and outline how the model could be extended to incorporate an evolving genetic architecture (L32–50 p20 (C and M)).

That said, we agree that the use of the term 'coevolutionary dynamics' could be misleading in this context, and we have now removed both instances from the manuscript.

When discussing implications, consider the possibility of neutral evolution for sex-specific traits. Under neutrality, a negative correlation between intersex correlation and divergence in sex-specific traits might be expected. Incorporating this scenario, perhaps with a simple simulation, would be beneficial for establishing a null expectation and is relevant to the present results and discussion.

We thank the reviewer for this thoughtful suggestion. We agree that neutral models can be informative under certain circumstances. However, in the context of our study, we do not consider neutrality to be an appropriate null model.

Under neutrality, the expected signed sexual dimorphism (i.e., the difference in trait means between the sexes) depends entirely on the initial conditions, as genetic drift has no directional tendency. While a negative correlation between intersex correlation and *absolute* sexual dimorphism might arise under neutrality, so might a positive one—the observed pattern would depend on the initial conditions. Moreover, we do not consider neutrality to be a meaningful paradigm in this context because—unless $r_{fm}=1$ —the expected absolute SD grows indefinitely over time. Under neutrality, provided the range of possible trait values that can be reached via mutation is broad, there is nothing constraining trait means to remain near any particular value. This is precisely why long-term morphological stasis is often interpreted as evidence for stabilizing selection in natural populations.

For this reason, we view stabilizing selection—not neutrality—as the more biologically appropriate model for studying sexual dimorphism in highly polygenic traits.

Minor:

Page 1 Line 10 adaption -> adaptation

Fig 3 Caption: dimporhism -> dimorphism

Long sentence Page 18 Line 10 - line 21. confusing.

We thank the reviewer for noticing the typos, which have both been corrected. In addition, the long sentence has been split into two sentences for clarity: L31-36 p18 (C and M).

Reviewer #2 :

This is my second review of the manuscript, and the authors have addressed my previous comments. In particular, the revised framing clarifies the limitations of characterizing the long-term relationship between the intersex correlation and sexual dimorphism under the assumption of a static genetic architecture. I have only minor comments remaining.

P1. Line 85, P2. 58-61. The evolution of genetic architecture is generally slower than the time scale of experimental evolution, but it's unclear why this would be especially relevant for male-female trait architectures compared to any two traits with shared genetic bases (e.g., highly allometric morphological traits). The cited articles do not clarify this distinction, and I'm not aware of any general intuition for it. While this is a minor point, it justifies the authors' decision to focus on non-evolving architectures for testing hypothesis 2, and greater clarity would be helpful.

We thank the reviewer for raising this point. Indeed, as we tried to clarify in this revised version, the fact that genetic architectures are generally—and also in the context of sexual dimorphism—expected to evolve slowly, beyond the time scale of most experimental evolution settings, is one of the reasons we chose to examine the relationship between sexual dimorphism and intersex correlation under a non-evolving genetic architecture.

While a detailed discussion of the evolvability of genetic architectures in general is beyond the scope of this study, we now include several references that explore the expected timescales and evolutionary dynamics of genetic architectures more broadly—including for allometric traits (Jones et al., 2003; Rajon & Plotkin, 2013; Jamamichi, 2022)—as well as work that jointly explore and compare the evolvability of G- and B-matrices (Barker et al., 2010). These are in L82-88 p2 (C) / L83-89 p2 (C).

P19, Lines 110 - 122. The authors may be interested in Carreira et al. 2009, which suggests that the genetic architecture of body size in *Drosophila melanogaster* can evolve greater sex-specificity via mutations at key developmental genes.

Carreira, V. P., Mensch, J., & Fanara, J. J. (2009). Body size in *Drosophila*: genetic architecture, allometries and sexual dimorphism. *Heredity*, 102(3), 246-256.

We appreciate this reference, which we were not aware of. We now include it when discussing mechanisms which can contribute to the change in genetic architectures, in L119 p19 (C and M).

Figure S4. The horizontal dashed line in part a is not specified in the caption.

Thank you for noticing. The caption for Figure S4 has been updated to specify the meaning of the dashed line.

August 7, 2025

RE: GENETICS-2025-308261R1

Dr. Laura Katharine Hayward
Institute of Science and Technology Austria
Evolutionary genetics
Am Campus 1
Klosterneuburg 1180
Austria

Dear Dr. Hayward:

Congratulations, your manuscript titled "The relationship between sexual dimorphism and intersex correlation: do models support intuition?" is accepted for publication in GENETICS! Many thanks for submitting your research to the journal, and for your work in response to the reviews.

To Proceed to Publication:

1. Format your article according to GENETICS style: <https://academic.oup.com/genetics/pages/author-guidelines>
2. Ensure that you comply with data and community resource citation guidelines:
<https://academic.oup.com/genetics/pages/author-guidelines#section-5-9-2>
3. Upload your final files at <https://genetics.msubmit.net>
4. Add oupsupport@scipris.com and genetics.oup@novatechset.com (or the domains @scipris.com and @novatechset.com) to your email program's "safe senders" list. You will be contacted by both at various points during the production process.

Notes:

- Your currently-accepted manuscript (unedited, as submitted, reviewed, and accepted) will be published at GENETICS and deposited into PubMed as an Advance Access article. Notify sourcefiles@thegsajournals.org before signing your license if you do not wish to publish your article via Advance Access.
- We invite you to submit an original color figure related to your paper for consideration as cover art. Please email your submission to the editorial office or upload it with your final files. You can submit a small-sized image for evaluation, and if selected, the final image must be a TIFF file 2513px wide by 3263px high (8.375 by 10.875 inches; resolution of 600ppi). Please avoid graphs and small type.
- After files are sent to Oxford University Press we use SciPris to manage article licensing and payment. If you do not have a SciPris account, you will receive an email from no-reply@scipris.com to sign up to use Oxford University Press' author portal. After logging in, follow the online instructions to sign your license and arrange any payment due.

If you have any questions or encounter any problems while uploading your accepted manuscript files, please email the editorial office at sourcefiles@thegsajournals.org.

Sincerely,

Graham Coop
Associate Editor
GENETICS

Approved by:
Sarah Otto
Senior Editor
GENETICS

Review comments (if applicable):